# Distinct immune cell infiltration patterns in pancreatic ductal adenocarcinoma (PDAC) exhibit divergent immune cell selection and immunosuppressive mechanisms

Pancreatic ductal adenocarcinoma has a dismal prognosis. A comprehensive analysis of single-cell multi-omic data from matched tumour-infiltrated CD45+ cells and peripheral blood in 12 patients, and two published datasets, reveals a complex immune infiltrate. Patients have either a myeloid-enriched or adaptive-enriched tumour microenvironment. Adaptive immune cell-enriched is intrinsically linked with highly distinct B and T cell clonal selection, diversification, and differentiation. Using TCR data, we see the largest clonal expansions in CD8 effector memory, senescent cells, and highly activated regulatory T cells which are induced within the tumour from naïve cells. We identify pathways that potentially lead to a suppressive microenvironment, including investigational targets TIGIT/PVR and SIRPA/CD47. Analysis of patients from the APACT clinical trial shows that myeloid enrichment had a shorter overall survival compared to those with adaptive cell enrichment. Strategies for rationale therapeutic development in this disease include boosting of B cell responses, targeting immunosuppressive macrophages, and specific Treg cell depletion approaches.

Pancreatic ductal adenocarcinoma (PDAC) has the worst survival of any common human cancer, with a 5-year survival of below 10%[1]. The mainstay of treatment is chemotherapy, however, approximately 15% of patients benefit from surgical resection, which can potentially provide cure in a subset of those patients. Despite the introduction of immunotherapy, the benefit in PDAC is minimal[2–6], and so there is an unmet need to develop better treatments.

Previous work from our group and others has suggested that there is a sizeable immune infiltrate in these tumours and understanding the nature of this infiltrate is critical for developing pragmatic immunotherapy strategies for PDAC[7–10]. We have previously shown that patients with high tumour lymphocyte infiltration at resection have a better prognosis than those that do not[8]. Furthermore, after characterising tumour infiltrating lymphocytes (TILs) in PDAC, we see that even though there is limited exhaustion in a subset of CD8 T cells,

we observed that a significant number of CD4 and CD8 T cells were senescent[7,8]. Additionally, we see activated Treg cells expressing checkpoints TIGIT, ICOS, CTLA4 and CD39[7,9] suggesting a strongly immunosuppressive microenvironment.

Other groups have made recent observations regarding the intra-tumoural immune infiltrate[11]. Peng et al. did the first large scale single cell analysis of PDAC and demonstrated that there was a complex immune infiltrate, and they highlighted that T cells were the dominant immune cell in the TME[10]. Steele et al. performed a second large single cell experiment and demonstrated that the predominant CD8 T cell exhaustion marker was TIGIT[12]. Schlack et al. have performed single cell sequencing with TCR sequencing. They have identified a heterogeneous lymphocyte infiltrate and trajectory analysis demonstrated similarities between inhibitory and dysfunctional populations[13]. Brouwer et al. had undertaken a single cell

✉ e-mail: s.sivakumar@bham.ac.uk; enas.abushah@path.ox.ac.uk; rachael.bashford-rogers@bioch.ox.ac.uk

cytometry by time of flight (CyTOF) analysis using a 41-marker panel focused on infiltrating lymphocytes. They found low levels of tissue resident cytotoxic CD8 T cells and they concurrently have low levels of PD1. Interestingly, the group has also found high levels of activated Treg cells and B cells[14]. Liudahl et al. used an immune focused multiplex IHC panel to evaluate leucocyte populations in a cohort of 135 PDAC patient samples. They demonstrated that the T cell to CD68 ratio is important in the treatment naive setting to demonstrate prognostic benefit[15].

There is a growing body of evidence describing a distinction between lymphocyte- and myeloid-enriched tumours and understanding what is driving this is critical to therapeutic interventions. Despite the growing number of datasets aimed at defining the nature of the different immune subsets within the PDAC TME, we still lack an understanding of the clonal evolution and differentiation pathways driving these populations. PDAC has traditionally been considered to have a low tumour mutation burden, suggesting a low prevalence of antigens to stimulate the immune response. However, seeing the presence of activated and exhausted cells within the TME, and associations between cytotoxic CD8 T cells, B cells and neoantigen quality with patient survival[16,17] suggests the presence of specific stimuli and warrants the investigation of the clonal distribution and evolution of both T and B cells. Multiple previous studies have shown that higher adaptive immune cell infiltration is associated with marginally better survival[8,18], however, both adaptive and myeloid enriched PDAC patients have dismal prognoses[6].

This study's objective is to elucidate the distinctive features of adaptive immune responses in patients' tumours with high levels of adaptive and myeloid cell populations and to identify the specific immune suppression pathways that set apart the myeloid-high and adaptive-high patient groups. To this end, we performed the largest and comprehensive analysis of PDAC-associated lymphocytes from tumour and blood to date using single cell multi-omics analysis coupled with the re-analysis of public PDAC scRNA-seq datasets[10,12]. We developed and applied single cell analyses to uncouple the distinct roles and contributions of different immune cell populations, the clonal nature across patient groups, the nature of immune cell migration and tissue adaptation, and provided insights into key pathways defining these differences. These insights will help in patient stratification and the development of personalised therapeutic approaches. The nature of B and T cells migrating between the tumour and draining lymph nodes is important for mounting effective anti-tumoural immune responses and establishing long-term systemic memory[19]. However, the signals responsible for B and T cell tumour infiltration, retention and egress, such as adhesion and chemokine milieu, are unclear. We aimed to explore the nature and determinants of B and T cell immunosurveillance in PDAC to identify pathways that can be targeted to improve immune cell trafficking. Finally, we introduce cutting-edge single-cell tools *scClonetoire* (which quantifies intra/inter subset clonality and other repertoire features), *scReptransition* (which measures clonal overlap within/ between samples), *SVMcelltransfer* (which annotates single cells in a scRNA-seq dataset confidently from a reference dataset containing multi-omics) and *scIsotyper* (which assigns high confidence VDJ chains per droplet). We then validated these findings at the protein level by using exploratory analysis of IHC measurements of a large cohort of primary resected, treatment-naïve clinical samples from the Adjuvant PAncreatic Cancer Trial (APACT[20]), an international phase 3 clinical trial of adjuvant chemotherapy, which validated the presence of these groups, and demonstrated their prognostic significance. This study lays the foundation for understanding why immunotherapy has so far not been successful in PDAC, and provides an avenue for identifying novel therapeutic targets based on an enhanced understanding of the patients' intra-tumoural immune composition.

## Results

### Single cell profiling of PDAC tumour immune cell infiltration across three datasets

To elucidate the heterogeneity of tumour immune cell infiltration, we performed single cell RNAseq (gene expression, GEX), ADT-seq (cell surface protein expression derived from Antibody-Derived Tags), B cell receptor (BCR) and T cell receptor (TCR) sequencing on CD45+ cells enriched from matched fresh tumour tissue following surgical resection of 12 treatment-naïve patients and matched PBMCs, herein referred to as PancrImmune. In addition to the PancrImmune dataset, we integrated and re-analysed the two existing PDAC scRNA-seq datasets from Peng[10] and Steele[12] (Fig. 1a and Supplementary Figs. 1–8). Integrative multi-omics analysis of GEX, ADT-seq and BCR/TCR-seq allowed for high confidence and quality annotations of B cell, T cell and myeloid populations (Fig. 1b and Supplementary Data 2), with robust detection of immune cell diversity. This high granularity analysis of immune cells in the blood and PDAC tumour infiltrate revealed a high complexity of immune cell infiltration in the TME with a wide variety of activated and regulatory immune cells in all major immune subsets (Supplemental Data 1 Figs. 1–5). The integrated Peng and Steele datasets only had GEX data and therefore we used a support vector machine (SVM) cell label transfer method, *SVMCellTransfer*, using the PancrImmune GEX data as a reference (Supplemental Data 1 Figs. 6–8, Supplementary Data 2, see methods). The resulting gene expression patterns of each cell annotation type reflected well the patterns observed in the PancrImmune reference dataset (Supplemental Data 1 Fig. 8).

### PDAC tumour myeloid infiltration positively associates with plasma cell abundance

Patients' tumours span a spectrum of immune cell infiltration, and higher intra-tumoural T cell frequencies are typically associated with improved patient survival (Supplementary Data 3). We therefore investigated next what might be mediating these differences. The proportion of immune cells consisting of tumour infiltrating myeloid cells inversely correlates with B and T cells consistently across datasets ($p$ values < 1e−7, Fig. 1c, $p$ values per dataset <0.018, Supplementary Data 4). To better understand the mechanisms underlying this, we compared patients with high B and T cell tumour infiltrate proportions (as a percentage of CD45+ cells), termed adaptive-enriched (AE) versus high myeloid, low B and T cell proportions, termed myeloid-enriched (ME) (Fig. 1d, e). This mirrors the prognostic signatures previously identified[8] and summarised in Supplementary Data 3. Indeed, the top 10 differentially expressed genes (DEGs) between groups (pseudobulk analysis) have predominantly B cell and T cell specificities for AE patients, or myeloid cell specificities for ME patients (Supplementary Data 5). The cellular subset proportions within B cells, T cells, NK cells and ILCs significantly differed between AE and ME groups across all three datasets and were clearly separable by PCA analysis (Fig. 1e and Supplementary Fig. 9a–c). Across all three datasets, we observed significantly reduced plasma cell abundance with increased overall B and T cell infiltration, along with consistent trends in proportions of other immune cell populations demonstrated across datasets (Fig. 1f and Supplementary Fig. 9d, e), suggesting that different mechanisms of differentiation, proliferation and recruitment may be acting in the different patient groups. Indeed, this is in agreement with previous studies shows an association between plasma cells and myeloid cells in lymphoid tissues[21–23].

We also confirmed that there are consistent significant differences in myeloid versus adaptive immune cell infiltration into the tumour when considering the total cell population (rather than just the proportion of CD45+ cells) in both the Peng and Steele datasets (Supplementary Fig. 9f). There were no significant differences in the proportions of non-immune cell types between the ME and AE patients, suggesting that tumour load (found in the epithelial cell

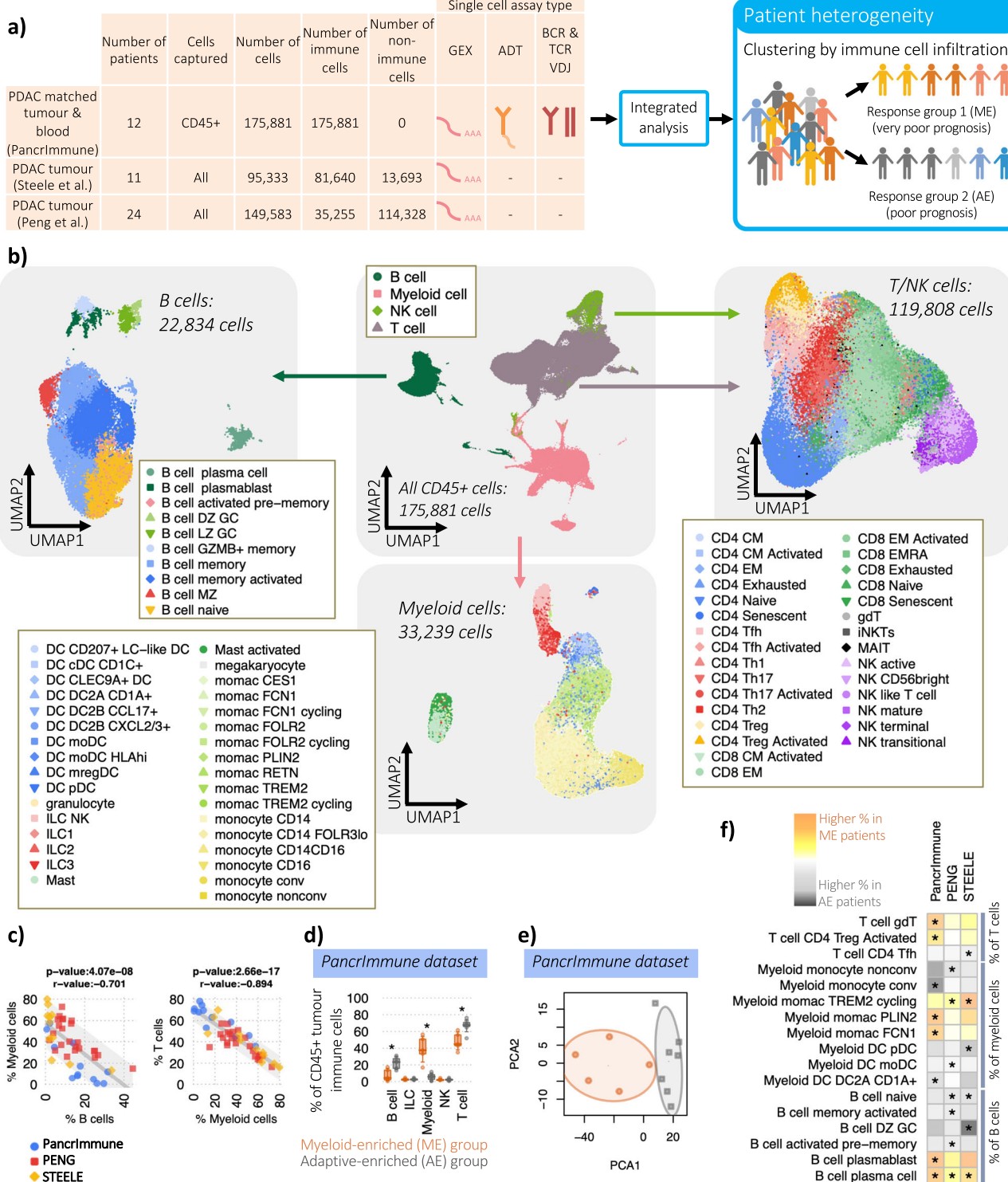

**Fig. 1 | Increased intra-tumoural lymphocyte infiltration is associated with distinct immune cellular compositions. a** Schematic of the datasets and analyses. Created in BioRender. (https://BioRender.com/y83y505). **b** UMAP dimensionality reduction of the intra-tumoural immune cells from the PancrImmune dataset depicting total immune cells (centre), B cells (left), myeloid cells (bottom) and T cells (right). **c** The correlation of (left) B cells and Myeloid cells and (right) myeloid cells and T cells as a proportion of total intra-tumoural immune cells across the three datasets, coloured blue, red and yellow for the PancrImmune (n = 12), Peng (n = 24) and Steele (n = 11) datasets respectively. R-values and p values calculated via Pearson linear regression, shaded area denotes 85th percentile confidence intervals. **d** The cellular proportions of the broad intra-tumoural immune cell types

between myeloid- (n = 5) and adaptive-enriched (n = 7) patients in the PancrImmune dataset. **e** Principal component analysis (PCA) of the intra-tumoural immune cell proportions per sample, coloured orange for myeloid-enriched (ME) patient samples and grey for adaptive immune cell enriched (AE) patient samples (PancrImmune dataset). Ellipses represent the 65th percentile intervals. **f** Heatmap of the differences in intra-tumoural cellular proportions between ME and AE patient tumour samples. The colour denotes the proportional skew between ME and AE patients. * denotes a significant difference between ME and AE patients (p value < 0.05). Statistical tests were performed by two-sided MANOVA. Boxplots define the 10th, 25th, 50th, 75th and 90th percentiles. Schematic created with BioRender.com.

compartment) and non-immune cell composition is not driving these differences. No correlations were observed with other patient factors including age, gender, prior disease history or diabetes status.

### Inverse correlation between ductal and immune cell subset proportions

Using the Peng and Steele datasets where all cells associated with the tumour were present, we were able to dissect the correlations between the immune and non-immune cell compartments. Although there was no correlation between the epithelial cells and any other non-immune cell type, we saw the strongest significant inverse correlations between epithelial (containing the tumour cells) with the myeloid, T cells and NK cell proportions (Supplementary Fig. 9f), suggesting direct or indirect immunosuppressive activity by the tumour cells. Moreover, there was no significant inverse correlation between the fibroblast compartment and the immune cell infiltration, which supports recent evidence disputing the previously held idea that the desmoplastic core limits immune infiltration[24].

### B cell selection is distinct between patient groups

Next, we examined the intra-tumoural B cells in greater detail given their divergent proportions between patient groups. The proportions of both plasma cells (PCs) and plasmablasts (PBs) of total B cells were significantly higher in ME tumours than AE tumours (Fig. 2a). IGHV gene usages and isotype features were distinct and clearly separable by PCA between AE and ME patients (Fig. 2b and Supplementary Fig. 10a, $p$ values < 0.0125) suggesting different B cell repertoire selection processes between patient groups[25].

### Reduced B cell class-switching in AE patients

Elevated IgA1 and IgA2 were observed in the intra-tumoural activated, memory, and antibody secreting PB cells in the ME patient group, whereas elevated IgG1 and IgM levels were observed in the AE group (Fig. 2c). These differential phenotypes are suggestive of differences in B cell signalling and germinal centre (GC) or tertiary lymphoid structure (TLS) responses. Interestingly, the patients associated with better prognosis (AE) had reduced class switch recombination (increased proportion of IgM B cells) compared to ME. Indeed, this observation of elevated IgM in the better-prognostic patients is supported when examining the larger PDAC TCGA dataset ($n$ = 67 patients at stage II), in which we identified a weak but significant association between both high IGHM with improved survival (Fig. 2d). In addition, IgM+ BCRs had lower levels of somatic hypermutation (SHM) (Supplementary Fig. 10b). Taken together the AE patients may have distinctive GC reaction outcomes. We note that the dominant *IGHA1*, *IGHA2* and *IGHM* isotype usages observed here also reflect what is seen in healthy pancreatic tissue (reanalysis of GTEx RNA-seq data, Supplementary Fig. 10c). This could suggest that the pancreatic environment and supporting draining lymph nodes preferentially support class-switching to *IGHA1/2* as observed in other GI tract locations, rather than a predominance of *IGHG1/2* observed in non-GI tract tissues[26]. These differences in isotype usages were not observed in blood (Supplementary Fig. 10d), in keeping with tissue-specific differences rather than differences in systemic B cell responses between patients. Fc receptors for IGHA (FCAR/CD89), which are known to have dual effect, either to inhibit or activate macrophage responses depending on either monovalent or multimeric IgA ligation[27], are predominantly expressed in pancreatic myeloid cells, and Fc receptors for IGHM (*FCMR*) are predominantly expressed in B, T and NK cells (Supplementary Fig. 10e). Together, this further strengthens the relationship between IGHM and improved survival (potentially as antigen presenting cells) as seen in lung cancer[28]. despite potentially reduced GC efficiency, as seen in the AE group, and the relationship between IGHA secretion and myeloid cell phenotypes driving one of the pathways of immune suppression in the ME patients.

### Increased GC B cell clonality in AE patients

We next assessed the clonality of the B cell subpopulations via two measures: intra-subset clonality which reflects specific cell populations which have undergone clonal expansion, and inter-subset clonality to reflect the expansion and differentiation between subpopulations (Fig. 2e). Intra-subset clonality quantifies the percentage of cells in clones of 2 or more cells per subset, measuring the clonality within the subset. Inter-subset clonality quantifies the percentage of cells of each cell type as members of clones of 3 or more cells across all populations, this indicates cells within each B cell subset that may be members of larger clones that span multiple phenotypes, reflecting B cell plasticity driven by the specific TME signals they encounter. In antibody-secreting cells (PCs and PBs) in ME patients, there is elevated clonality suggesting that these are arising from recent or ongoing immune reactions in ME patients. The relatively higher levels of inter-subset clonality in the naïve B cells in ME patients, despite the expectedly low intra-subset clonality, is likely driven by the activation and clonal expansion of some naïve B cells and transition to other B cell phenotypes. The highest intra-subset clonality was observed in the GC B cells in keeping with these cells partaking in clonal B cell responses, potentially in TLSs. Indeed, these comprised the largest proportion of B cells from expanded clones (inter-subset clonality) along with memory B cells in the AE group only, with significantly lower inter-subset clonality in these cells in the ME patients. There were no significant differences in the proportions of GC B cells, which may be due to the small numbers of patients and GC cells. Together, this is suggestive of GC formation in AE patients with greater clonal expansion in GC B cells, however, resulting in unswitched memory B cells rather than intra-tumoural plasma cells. GC B cells in ME patients are not as clonal, however, the responses are predominantly *IGHA1* and *IGHA2*, and are more likely to differentiate into plasma cells.

### B cells comprise a major pool of antigen presenting cells in AE patients

B cells are considered one of the major professional antigen-presenting cells (pAPCs) via the MHC II pathway[29], however their role in the activation of T cells in PDAC has not been fully explored. Here, we derived a pAPC score for each cell by quantifying the feature expression programme for MHC II and accessory pathway molecules (see Methods). We defined pAPCs as those above a threshold derived from the optimal separation between scores from dendritic cells (DCs, known pAPCs) and CD8 T cells (known non-pAPCs) (Fig. 2f). Together with DCs, >65% of naive and antigen-experienced (activated and memory) B cells and monocyte-derived macrophages (MoMacs) are pAPCs (Fig. 2g). Whilst a mean of 57.6% of pAPC are MoMacs, and 21.1% of pAPC are DCs in the ME patients, only 16.0% of pAPCs are B cells (Fig. 2g). Interestingly B cells comprise a major source of antigen presenting cells in the AE patient group, 80.4% of pAPCs are B cells ($p$ values < 0.05), mainly antigen-experienced (activated and memory) B cells. This trend is validated in the Peng et al. dataset (Supplementary Fig. 10f–i, $p$ values < 0.05). Given the elevated T cell infiltration in the AE patients (Fig. 1f) and the significant contribution of B cells to the pAPC pool, this highlights a potential role for B cells in PDAC TME in shaping T cell activation.

### Increased CD8 T cell clonality in AE patients

To understand the drivers behind the increased T cell prevalence in the AE patients, we performed clonality analysis of the T cell populations using the TCR sequencing data from intra-tumoural T cells in the PancrImmune dataset. We observe an increased CD8 T cell clonality in the AE group (Fig. 3a, $p$ values < 0.05) which suggests that the increased PDAC lymphocyte infiltration is partly driven by clonal activation and expansion. This was observed in both tumour-infiltrating and peripheral blood T cells, implying that some of the tumour expanded clones could potentially be in the blood. We further

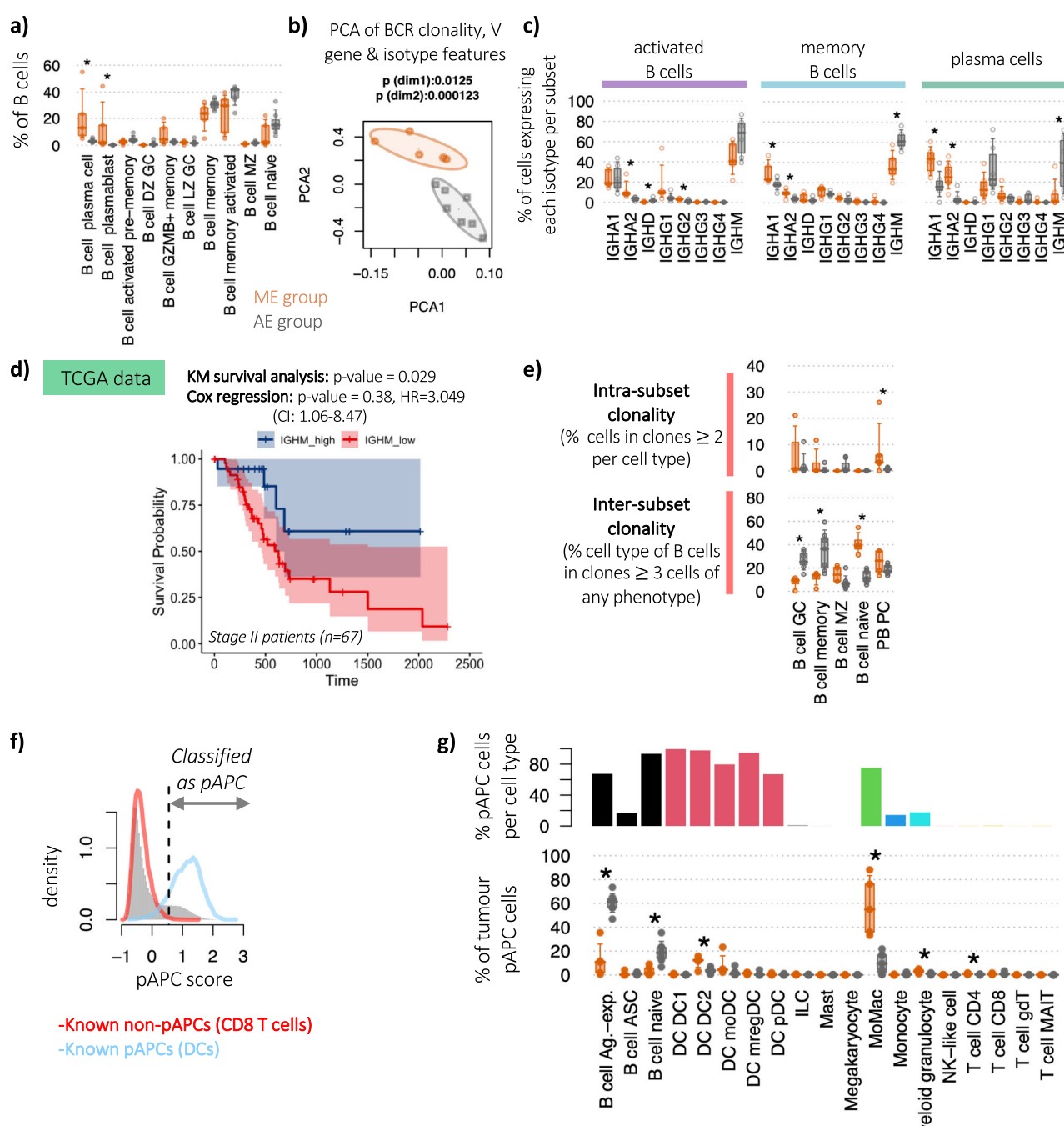

**Fig. 2 | Increased PDAC lymphocyte infiltration is associated with differences in B cell selection, clonal expansion and class-switch recombination. a** Immune cell subset proportions between ME and AE patient groups within tumour B cell subsets as a proportion of total B cells (orange represents ME patients and grey represents AE patients) within the PancrImmune dataset. **b** Principal component analysis (PCA) of the tumour BCR clonality, IGHV gene usages and isotype usages, coloured by patient group. **c** The proportions of tumour B cells within activated, memory and plasma cells expressing each isotype, coloured by patient group. **d** Survival plot for high vs. low IGHM expression with a *p* value for Kaplan–Meier (KM) plot (log-rank test) and Cox proportional hazards model (two-sided Wald test). HR hazard ratio, CI confidence interval. **e** Clonality of the tumour B cell subpopulations between the ME and AE patient groups via two measures: (top) *intra-subset clonality* (the percentage of cells in clones >2 cells per subset, measuring the clonality within the subset thus reflecting specific cell populations which

are actively expanding), and (bottom) *inter-subset clonality* (the percentage of cells of each cell type as members of clones >3 cells across all populations, demonstrating, this indicates cells within each B cell subset that may be members of larger clones than span multiple phenotypes, reflecting B cell plasticity of expanding clones). **f** Histogram of the professional antigen presentation (pAPC) scores for (grey) all tumour cells, (red) tumour CD8 T cells and (blue) tumour DCs. Dashed line indicates the threshold for classification of pAPCs. **g** (top) Bar chart of the percentages of pAPCs comprising each cell type, and (bottom) the proportion of tumour pAPCs comprising each cell type between patient groups. All single cell analyses in this figure were performed on the intra-tumoural B cells from PancrImmune dataset. * denotes *p* values < 0.05, and tests were performed by two-sided MANOVA. ME patients have an *n* = 5 and AE patients have an *n* = 7. Boxplots define the 10th, 25th, 50th, 75th and 90th percentiles.

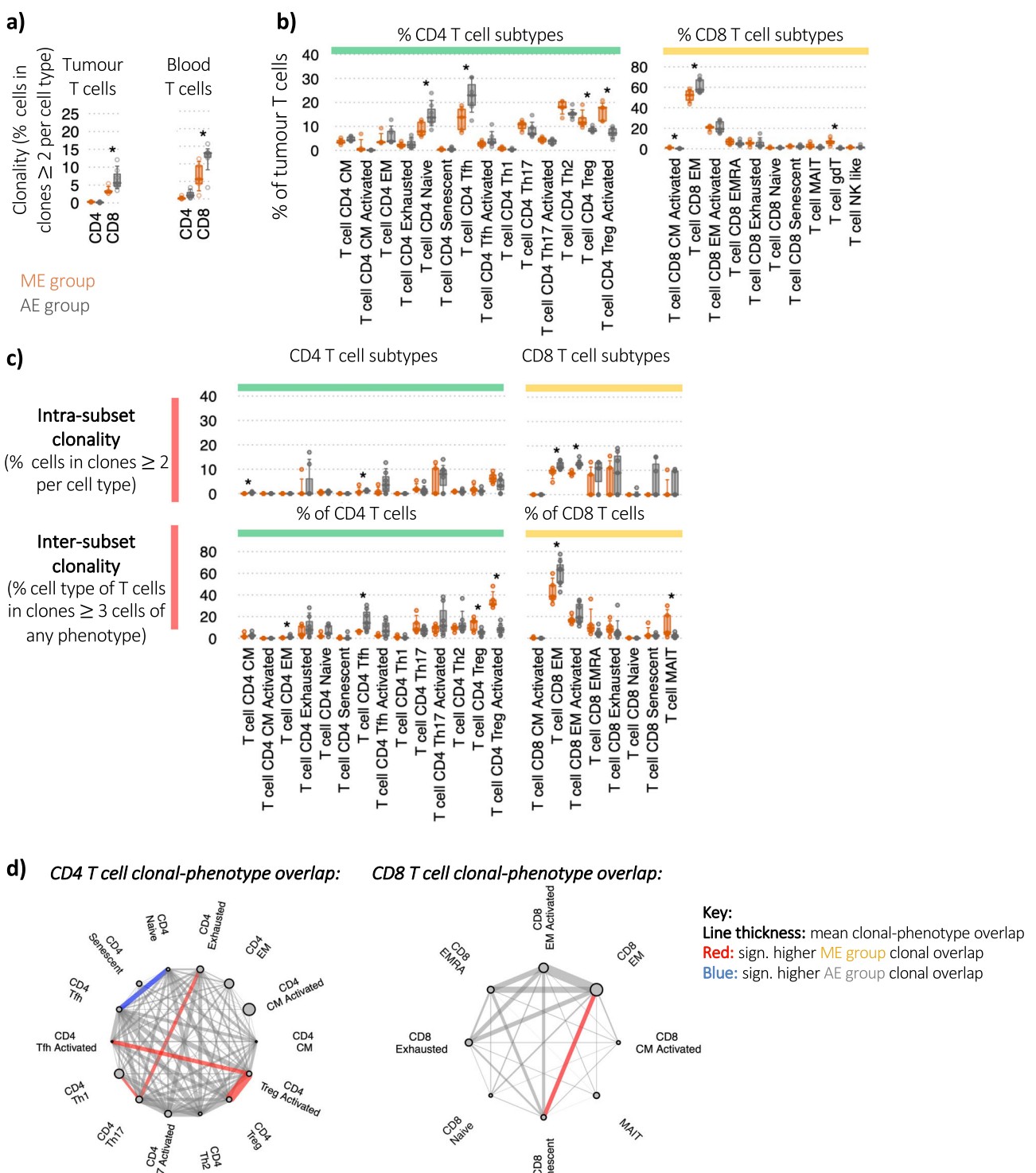

analysed the different sub-populations for any differential presence of T cell subsets between the two groups of patients. In addition to the elevated CD4:CD8 T cell ratio in ME patients (Supplementary Fig. 9a, *p* value < 0.05), ME tumours showed higher proportions of Treg, activated Treg, and gamma-delta (gd) T cells, while AE tumours had higher proportions of T follicular helper (fh), naïve CD4 cells and CD8 effector memory (EM) cells (Fig. 3b). The increased proportion of CD8 EM cells in the AE group further reinforces the role of clonal activation in driving T cell infiltration, and the elevated proportions of Tfh cells in the AE patients further supports the inter-relationship between B and

T cells in PDAC. By contrast, the elevated proportions of Treg cells and activated Treg cells in the ME group could reflect the increased immunosuppressive TME of these tumours. A validation was performed via cellular deconvolution of the PAAD TCGA dataset (*n* = 156 patients) showing that indeed Treg proportions (as a proportion of total T cells) correlated with myeloid cell proportions (as a proportion of total immune cells) (Supplementary Fig. 11a), whereas the T cell proportion of total immune cells inversely correlated with the proportion of myeloid cells (Supplementary Fig. 11a). Furthermore, TCR clonality and TRVB features are distinct between AE and ME patients

**Fig. 3 | Increased PDAC lymphocyte infiltration is associated with increased activated Treg clonality. a** The clonality of intra-tumoural T cells within total CD4 and CD8 T cell populations, measured by percentage of clones consisting of >2 cells. Orange represents ME patients and grey represents AE patients. **b** Immune cell subset proportions between ME and AE patient groups within tumour T cell subsets as a proportion of total CD4 and CD8 T cells, respectively. **c** Clonality of the tumour T cell subpopulations between the ME and AE patient groups via two measures: (top) *intra-subset clonality* (the percentage of cells in clones >2 cells per subset, measuring the clonality within the subset thus reflecting specific cell populations which are actively expanding), and (bottom) *inter-subset clonality* (the percentage of cells of each cell type as members of clones >3 cells across all populations, demonstrating, this indicates cells within each T cell subset that may be members of larger clones than span multiple phenotypes, reflecting T cell plasticity of expanding clones). **d** Level of tumour TCR clonal sharing between (left) CD4 T cell and (right) CD8 T cell populations. Each line represents a sharing of TCR clones between cell types, and the line thickness denotes the mean relative level of sharing. A red line denotes that the clonal sharing between the corresponding cell types is significantly higher in the ME patients than AE, and a blue line denotes that the clonal sharing between the corresponding cell types is significantly lower in the ME patients than AE. The size of the dot represents the mean relative frequency of the corresponding cell type. All analyses in this figure were performed on the intra-tumoural B cells from PancrImmune dataset. * denotes *p* values < 0.05, and tests were performed by two-sided MANOVA. ME patients have an *n* = 5 and AE patients have an *n* = 7. Boxplots define the 10th, 25th, 50th, 75th and 90th percentiles.

(Supplementary Fig. 11b: PCA plot of T cell repertoire features, Supplementary Fig. 11c: V gene usages). Together, this is suggestive of different T cell selection processes between ME and AE patients.

## Activated Treg cells are enriched in expanded T cell clones

We next assessed the clonality of the T cell subpopulations via intra-subset clonality and inter-subset clonality (Fig. 3c). Intra-subset clonality measures the clonality within the subset, and inter-subset clonality quantifies the cells within each T cell subset that may be members of larger clones that span multiple phenotypes, reflecting T cell plasticity driven by the specific TME signals they encounter. The highest intra-subset clonality was observed in the CD8 EM, activated EM, effector memory cells re-expressing CD45RA (EMRA), exhausted and senescent T cells. There was preferential intra-subset expansion in the AE group of the CD8 EM and CD8 activated EM T cells. Taken with the increased percentages we observed earlier (Fig. 3b), this provides additional support that the increased CD8 EM presence in AE patients is driven by local expansion within the tumour. Notably, although activated Treg cells, which are marked by the highest expression of immunomodulatory molecules *TIGIT*, *ICOS*, and *CTLA4*, as well as the transcription factors *FOXP3* and *IKZF2*, were not the most clonal CD4 T cell population (intra-subset clonality), which is to be expected from a polyclonal regulatory T cell population, they were members of the most expanded clones (inter-subset clonality, Fig. 3c), which was significantly more expanded in the ME patient group, suggesting some of the observed Treg cells could be differentiating from other CD4 T cells within the TME.

## Distinct T cell clonal fate between AE and ME patient groups

Through quantifying the relative overlap of clones between different phenotypes within the CD4 and CD8 T cell populations, lineage patterns can be discerned (Fig. 3e) over the history of the tumour prior to sampling. Indeed, the tumour aetiology in PDAC (and most cancers) has been shown to occur over years or decades[30], and clonal sharing analysis can determine how cells from the same clone (and thus antigen specificity) have differentiated within the whole history of each clone. Indeed, highest clonal overlap in the CD8 T cell populations was observed between CD8 EM, activated EM and CD8 senescent T cell subsets, suggesting that common antigens are driving the expansions across these populations. Elevated intra-subset clonality was observed in the ME patients between CD8 EM and CD8 senescent T cell populations in the tumour, suggesting that activated T cells are pushed to dysfunctional phenotypes. The development of senescence in both patient groups suggest that the TME is conducive to the generation of these populations through potentially shared pathways. In the CD4 T cells, activated Treg cells have the highest degree of clonal overlap with activated Tfh and activated Treg populations, which was significantly higher in the ME patients. In AE tumours there was higher clonal overlap between CD4 naïve and Tfh T cells. Taken together, these results point to differential CD4 polarisation of intra-tumoural CD4 T cells between the ME and AE groups.

## T cell clonality between tumour and blood are distinct

Using the matched blood and tumour samples, we observed that the clonality of T cells between blood and tumour is highly divergent (Supplementary Fig. 11d). Within the blood CD4 T cell compartment, only CD4 senescent and Th1 T cells had high levels of intra-subset clonality. We noted that Treg cells were not clonal in blood, with <5% of these cells comprising expanded clones, a possible indication that Treg cells from TME expanded clones are tissue resident. In the CD8 T cell compartment, the CD8 EMRA and senescent populations were the most clonal populations and are significantly more clonal than their corresponding tumour T cell counterparts (Supplementary Fig. 11e); which is to be expected for these populations as they are driven through chronic antigen exposure (in many cases viral)[31]. There was no difference in CD8 T cell clonality observed in the blood between ME and AE patients (Supplementary Fig. 11d). Finally, to determine if the T cell responses within the tumour were enriched for systemic anti-viral responses rather than potential tumour-specific responses, we screened the tumour and blood-derived TCRs against a library of known anti-viral TCRs (see 'Methods'). Indeed, we found that anti-viral T cell clones are not enriched in the tumour compared to the blood and there were no widespread consistent differences between ME and AE patients (Supplementary Fig. 11f). This supports that tumour clones are not detectably enriched for systemic non-tumour-reactive clones. However, we note that as these patients will have diverse HLAs and so T cell responses may not be fully represented in published databases of known TCR antigen specificity.

## Immunosurveilling and resident tumour-infiltrating B and T cell clones are phenotypically distinct

Our dataset benefits from having matched blood and tumour samples taken at the same timepoints allowing us to perform analysis to identify circulating tumour infiltrating lymphocytes (TILs). These can be identified from clones shared between blood and tumour and represent clonally expanded lymphocytes recirculating between tumour and blood and will therefore be critical for immunosurveillance. We identified B and T cells clones that were (1) shared between blood and tumour (recirculating clones), (2) tumour-only (non-circulating TIL clones) and (3) blood-only clones. These states, by definition, exhibit different cellular tendencies for tumour ingress and/or adhesion (Fig. 4a and Supplementary Data 6). Circulating TIL B cells were enriched for naïve B cells in both AE and ME groups, suggesting that naïve B cells may be major components of immunosurveillance and selected B cells become activated in response to the TME (Fig. 4b). This is supported by the elevated *IGHM* usage in circulating TIL B cell clones (Supplementary Fig. 12a). Non-circulating TIL B cells were enriched for antibody secreting B cells and activated memory, suggesting that these are much less mobile upon tumour entry or differentiation. The dynamics of immune cell infiltration is explored in the next section.

Next, we explored the dynamics of immune cell infiltration. Whilst there was no significant difference of specific CD4 T cell subsets between circulating and non-circulating TILs, circulating CD4 TILs are

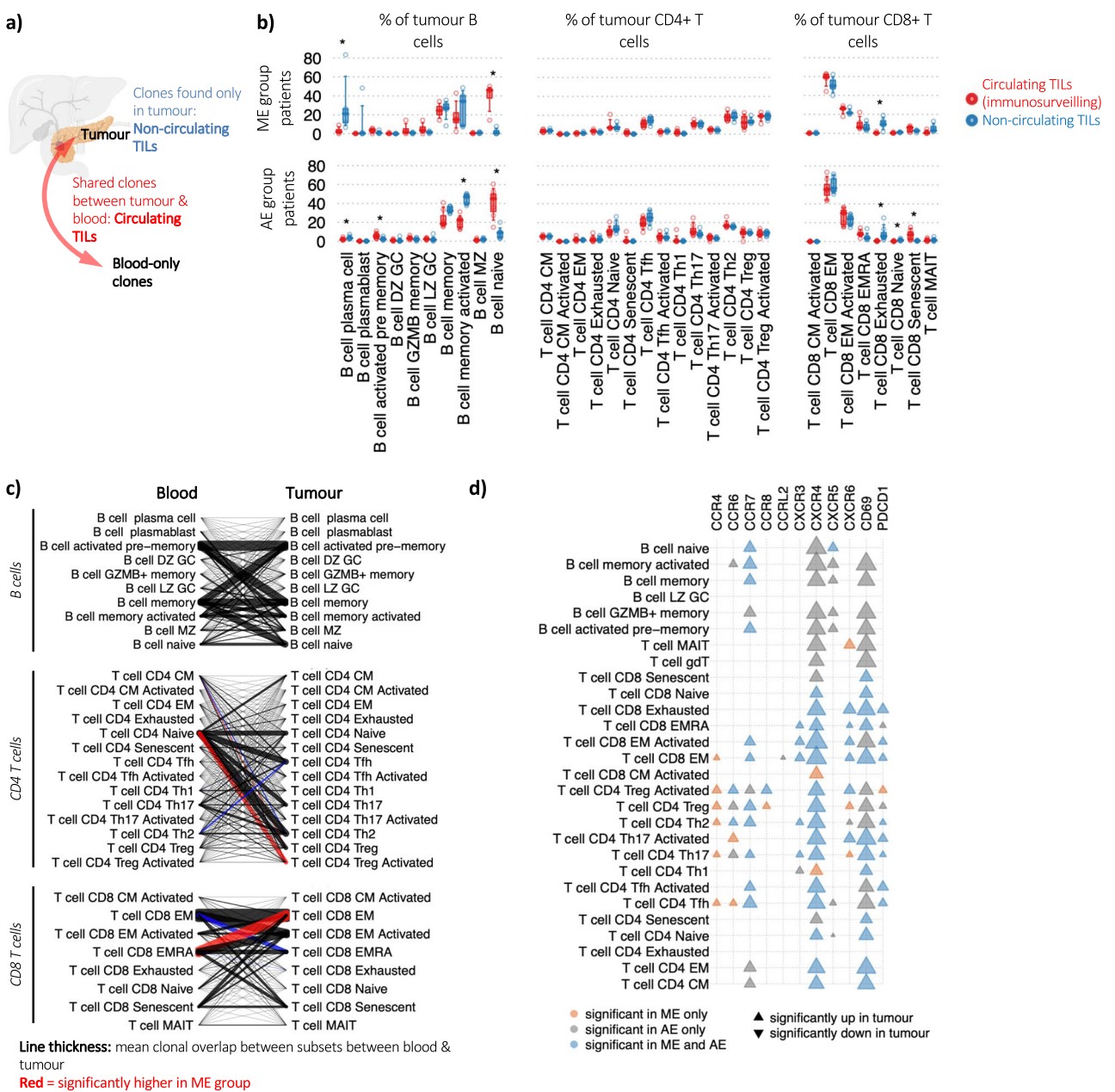

**Fig. 4 | Immunosurveilling and resident B and T cell clones are phenotypically distinct. a** Schematic of clonal definitions: B and T cells clones that are **a** shared between blood and tumour (recirculating clones), **b** tumour-only (non-circulating TIL clones) and **c** blood-only clones. Created in BioRender. (https://BioRender.com/y83y5o5). **b** The percentage of tumour B cells, CD4 T cells, and CD8 T cells that (red) have clonal members in the blood or (blue) no clonal members in the blood for the ME patients (top) and AE patients (bottom). **c** Clonal migration overlap plot, showing the linked phenotypes between blood and tumour B and T cells from shared clones between sites. Line thickness represents the relative means calculated over each patient. Red lines indicate that the corresponding clonal sharing between the corresponding cell types is significantly higher in the ME patients than AE, and a blue line denotes that the clonal sharing between the corresponding cell

types is significantly lower in the ME patients than AE. **d** Heatmap of DGE between blood and tumour biopsy between ME and AE patients per lymphocyte cell type. For each chemokine receptor and for each cell type, the upwards triangle denotes significant elevation of expression in tumour compared to blood and downwards triangle denotes significant reduction of expression in tumour compared to blood. The triangles are coloured orange, grey and blue if the significance is observed in ME patients only, AE patients only or both, respectively. The sizes of the triangles denotes relative mean expression. All analyses in this figure were performed on the PancrImmune dataset using both the blood and tumour samples. * denotes *p* values < 0.05, and tests were performed by two-sided MANOVA. ME patients have an *n* = 5 and AE patients have an *n* = 7. Boxplots define the 10th, 25th, 50th, 75th and 90th percentiles.

dominated by Treg cells, Tfh, and Th2 (Fig. 4b). Circulating CD8 TILs are dominated by CD8 EM T cells, which is consistent with the arrival of activated CD8 T cells from the tumour-draining lymph nodes. However, these were not statistically enriched compared to non-circulating CD8 T cell clones only found in the tumour. As expected, exhausted

clones were enriched in the TME where they are most likely to encounter their antigen.

We did not observe differences in the proportions of total recirculating B and T cell TIL clones between ME and AE patients (Supplementary Fig. 13b). However, recirculating B and T cell TIL clones were

significantly more expanded than clones private to the tumour or blood clones (Supplementary Fig. 13c). Finally, through screening the TCRs with a library of known anti-viral TCRs, we found that anti-viral T cell clones were not enriched in the circulating TILs compared to the non-circulating and blood-only T cell clones (Supplementary Fig. 13d). This supports the conclusion that recirculating TIL clones are not enriched for systemic non-tumour-reactive clones.

### Dynamics of recirculating tumour-infiltrated B and T cells

Next, we considered how the phenotype of clonally-related B and T cells differ between the blood and the tumour. This can be measured through determining the phenotypes of B and T cells within the same clone shared between the blood and tumour (Fig. 4c and Supplementary Fig. 13e–g). Many of the recirculating B and T cells have different phenotypes between blood and tumour, suggesting extensive intra-tumoural B and T cell differentiation within the tumour site and/or distal from the tumour. For B cells, the majority of recirculating B cells are derived from tumour-infiltrated naive, memory and activated memory B cells. This suggests that selected naive B cells from the blood infiltrate the tumour, and these differentiate to express memory B cell markers before recirculating. This also provides evidence that the tumour may be a major site of B cell activation.

For CD4 T cells, the largest overlap occurs between CD4 naïve and T helper phenotypes (Tfh, Th17, Th2, and Treg cells) (Supplementary Fig. 13f), suggesting naïve CD4 cells are being polarised based on intra-tumoural factors. The ME patient group has significantly higher overlap between naïve CD4s and activated Treg cells, supporting that the myeloid-enriched TME in these patients is driving the differentiation and proliferation of activated Treg cells from naïve cells. Blood CD8 senescent are predominantly related to CD8 EM, activated EM, EMRA and senescent, suggesting that these cells are derived from highly activated effector T cell clones, as expected[32]. Indeed, the clonal relatedness of blood CD8 EMRA and tumour CD8 EM T cells is supported by the observation that these subsets are the most clonal populations in the blood and tumour CD8 populations, respectively (Supplementary Fig. 10h). Overall, these results demonstrate that the TME can differentially shape the B and T differentiation in the two patient groups.

### B and T cell infiltration is dependent on chemokine receptor upregulation

Previous reports have shown that chemokines are critical for the infiltration and egress of immune cells from tumours, including the CXCR4-CXCL12 axis shown in mouse models of melanoma[33], as well as a pre-requisite for the formation of TLSs, including the CXCR5-CXCL13 axis[34]. Therefore, we considered the expression of key lymphocyte chemokine receptors upon infiltrating into the tumour which is possible to assess between matched tumour and blood samples in the PancrImmune dataset where this is possible. The chemokine receptors *CCR6*, *CCR7*, *CXCR3*, *CXCR4*, *CXCR5*, and *CXCR6* have the highest expression across lymphocytes (Supplementary Fig. 14a), and *CCR8*, a known hallmark of tumour infiltrating Treg cells, is exclusively expressed on Treg cells[35]. We observe significant correlations between some chemokine receptors and lymphocyte infiltration, including *CCR8* expression correlating with activated Treg levels (Supplementary Fig. 14b). Differential gene expression (DGE) between blood and tumour infiltrating B and T cell subsets (see 'Methods') revealed that multiple chemokines and their receptors are upregulated upon entry to the tumour (Fig. 4d and Supplementary Fig. 14c). Upregulation of chemokine receptors in TILs implies that they are central to recruitment and maintenance of these immune cell types within the tumour. Of note, *CXCR4* was significantly upregulated across 24 out of 28 lymphocyte populations in the AE patient group, but in the ME group, *CXCR4* was not upregulated in B cells, MAIT, gamma/delta or CD8/CD4 senescent T cells, in accordance with their lower prevalence in this

patient group. Similarly, CXCR5 was only observed to be upregulated in tumour non-naive B cells and Tfh T cells only in the AE but not in ME patients. Lower *CXCR5* in tumour B cells and Tfh T cells in ME patients will likely impact the effectiveness of B cell migration, retention and responses within the tumour site. *CCR8* expression is significantly increased in intra-tumoral Treg cells compared to blood, predominantly in ME patients, with the highest *CCR8* expression observed in activated Treg cells. Indeed, the same trends were observed when considering only immunosurveilling clones (clones shared between blood and tumour) (Supplementary Fig. 14d, e). Overall, we observed reduced chemokine receptor expression in intra-tumoural ME patient lymphocytes.

Finally, we show that only TIL T cell clones tend to be acutely activated with elevated *CD69* and *PDCD1* (PD1) compared to the blood (Fig. 4d). However, the significant upregulation of *CD69* was observed in only AE patients for B cells and multiple CD4 T cell populations, suggesting reduced activation in specific lymphocyte subsets in ME patients.

### Myeloid cells in ME patients dominate cell-cell communication

We have thus far shown that differential immune cell subtype frequencies distinguish ME and AE patients and lymphocyte-associated differences. We next examined the cell-intrinsic differences in cell-cell communication between immune cells between ME and AE patients. Here we considered cell-cell interaction strengths between known cytokine- and inflammation-associated ligands and their receptors (see 'Methods'). Unlike in existing methods[36,37], we developed a workflow that considered only immune cell receptor-ligands, allowed for normalisation between samples, and considered cell-intrinsic differences only, rather than incorporating cell number. The signalling strengths between each pair of cell subtypes for each receptor-ligand pair was calculated by multiplying the percentage of cells per cell subtype expressing each respective gene for each patient. Thus, the strengths are independent of the total proportions of each cell type within the tumour (Fig. 5a and Supplementary Data 8). The cell-cell communication network depicted an expected high level of complexity within the tumour microenvironment with each cell subtype providing and receiving signals from many other cells. However, within this complexity, several features were clearly observed. Firstly, ME patients had significantly higher levels of signalling between myeloid and T cell populations, and AE patients had higher levels of signalling between B cell and T cell populations. Indeed, enumerating the number of incoming and outgoing interactions (corresponding to or from cell-surface receptors, respectively, Fig. 5b) clearly demonstrated that immune signalling within the tumour was dominated by myeloid cells in ME patients and B and T cells in AE patients. The highest levels of incoming and outgoing interactions within ME patients were from MoMac, moDC and granulocyte populations, whereas the highest levels within AE patients were from CD8 EM T cells and memory B cells. Although the proportions of GC and MZ B cells were very low (<5% of total B cells, Fig. 2a), we observed that these cells have considerable contributions to cell-cell signalling, and indeed significantly higher interactions were seen in the AE patients.

The top significantly enriched immune modulator in the ME patients was *SPP1* (Fig. 5c) which encodes for osteopontin, and is overexpressed in PDAC and known to potentiate tumour cell stemness, M2 macrophage polarisation[38], checkpoint expression[39] and is associated with poorer survival across multiple cancers including PDAC[40]. *AXL* was the third most ME-enriched cytokine that induces mregDC formation and upregulation of PD-L1 expression[41]. Indeed, the top 30 significantly enriched immune modulators in ME patients included *CCL8*, a ligand for the tumour infiltrating Treg cells chemokine receptor *CCR8*, *PVR*, a ligand for the T cell checkpoint protein TIGIT and *ITGA8*, which is known to activate TGFb (Fig. 5c). Nine significantly enriched immune modulators (*p* values < 0.05) are known

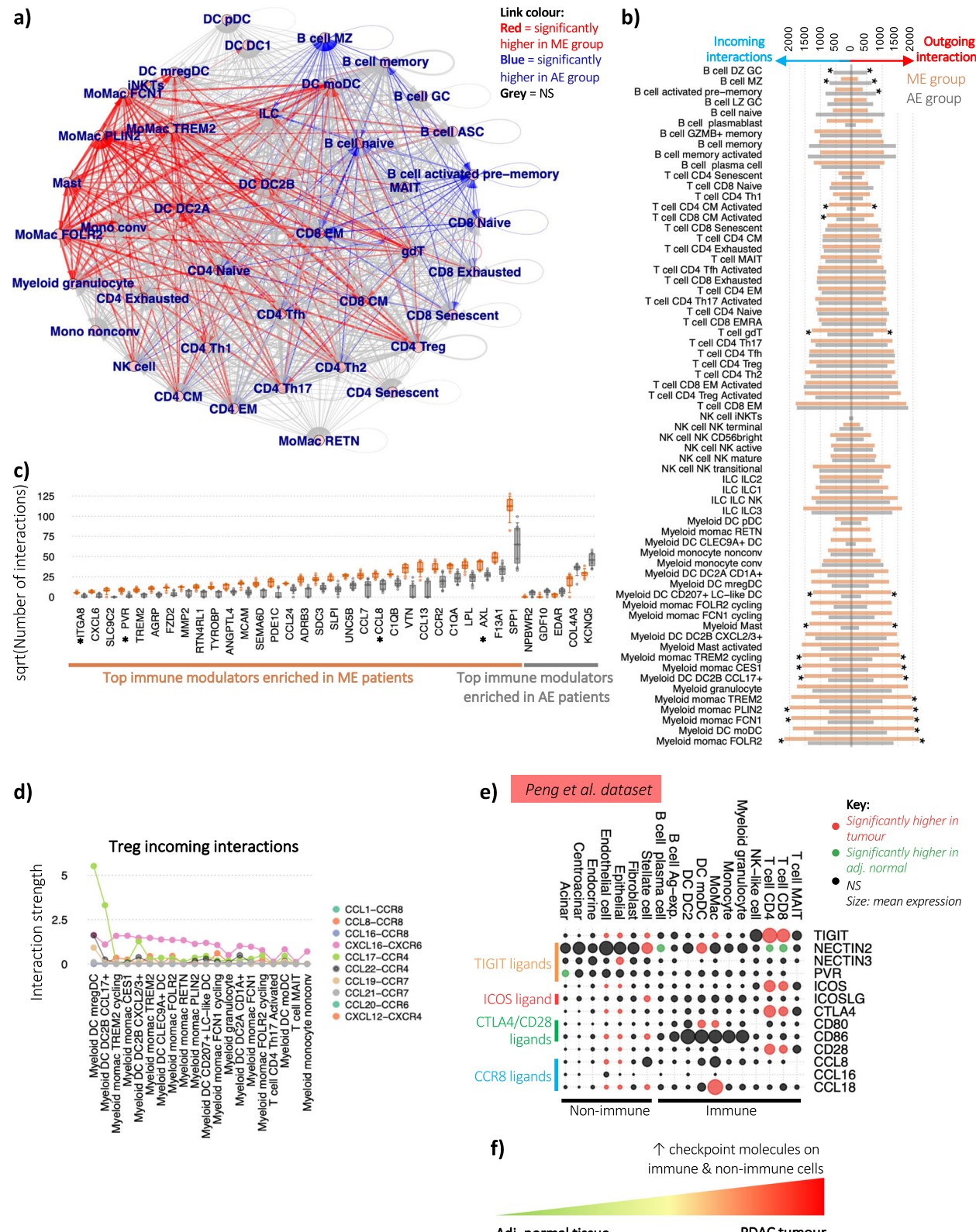

chemokines associated with immunosuppression or apoptosis (Supplementary Fig. 15a). Of note, both the inhibitory protein SIRPA and its receptor, CD47, are elevated in ME patients.

We next considered the signalling interactions to Treg cells which are known for being associated with immunosuppression within the tumour microenvironment. The ranked interaction strengths between the key Treg chemokine receptors (*CCR4*, *CCR8*, *CXCR4* and *CXCR6*)

and their ligands per cell type (,) showed that the incoming interactions with Treg cells were dominated by myeloid cells, notably mregDCs (driven by their expression of *CCL17* which interacts with CCR4 from Treg cells) as well as the regulatory axis CCL22-CCR4 which promotes Treg function[42], and DC2B CCL17+ and multiple MoMac populations (driven by their expression of *CXCL16* which interacts with CXCR6 from Treg cells). Finally, interactions with the Treg-exclusive

**Fig. 5 | Distinct regulatory mechanisms between patients with different immune cell infiltration. a** Intercellular immune modulator communication network between intra-tumoural immune cells, where each line thickness corresponds to the mean number of receptor-ligand interactions between the corresponding pair of cell types. A red line denotes that the number of receptor ligand-interactions between the corresponding cell types is significantly higher in the ME patients than AE, and a blue line denotes that the number of receptor ligand-interactions between the corresponding cell types is significantly lower in the ME patients than AE. **b** Quantification of the number of incoming and outgoing interactions per cell type split by ME and AE patient groups, calculated as a sum of all receptor-ligand pairs identified between all cell types. Bars indicate the means for each patient group, and * denotes p values < 0.05 between groups. **c** The number of interactions of the top 30 significantly enriched immune modulators in ME patients and all the top 30 significantly enriched cytokines, chemokines and immune-modulators in AE patients (p values < 0.05). **d** The top 20 ranked interaction strengths between the

key tumour Treg receptors (CCR4, CCR8, CXCR4 and CXCR6) and their ligands per cell type, coloured by receptor-ligand interaction type. **e** Differential checkpoint gene expression between adjacent normal pancreatic tissue and PDAC in both immune and non-immune cell compartments (using the Peng et al. dataset). Red circles indicate significantly higher expression in the tumour and green circles indicate significantly higher expression in the adjacent normal pancreatic tissue. Circle size indicates relative mean gene expression per cell type. **f** Schematic of the checkpoint expression landscape between healthy and pancreatic tumour tissue. All analyses in this figure were performed on the intra-tumoural immune cells from the PancrImmune dataset unless otherwise indicated. * denotes p values < 0.05, and tests were performed by two-sided MANOVA. For the PancrImmune dataset, ME patients have an $n = 5$ and AE patients have an $n = 7$. The Peng et al. data had 11 samples from adjacent normal and 24 samples from PDAC tissue. Boxplots define the 10th, 25th, 50th, 75th and 90th percentiles.

receptor CCR8 were dominated by MoMac expression of *CCL8*. Indeed, MoMacs were more numerous in ME patients (Supplementary Fig. 8a), and thus would support the infiltration of Treg cells into the tumour region. Whilst the expression of the Treg-associated chemokines, *CCL8* and *CCL16*, was observed in non-immune cells, including epithelial cells (which includes the tumour cells) (Supplementary Fig. 14d), the highest expression of these chemokines was within the myeloid MoMac and DC populations. Together, these findings demonstrate a key role of myeloid cells in promoting the immune-regulatory nature of the PDAC TME.

### Checkpoint genes are upregulated across both immune and non-immune cell subsets in PDAC

Finally, we examined the immunosuppressive nature of the whole TME including non-immune cells. Differential gene expression (DGE) analysis between PDAC and normal adjacent tissue from the Peng et al. dataset showed significantly elevated checkpoint gene expression in both immune and non-immune cell compartments (Fig. 5e and Supplementary Fig. 14e). While T cells are the primary source of *TIGIT* expression, stellate, epithelial and endothelial cells also have increased expression in the tumour compared to pancreatic adjacent normal tissue. Likewise, *ICOS* and *CTLA4* are primarily expressed by T cells, but are significantly higher in expression in tumour compared to pancreatic adjacent normal tissue in epithelial and endothelial cells. Differential expression of TIGIT, ICOS and *CTLA4* ligands were observed in both immune and non-immune cell types. Treg-associated chemokine receptor *CCR8* ligands, *CCL8*, *CCL16* and *CCL18*, were also elevated in tumour tissue stellate, epithelial and endothelial cells. Although the highest levels of *CCL18* was expressed by MoMacs, stellate cells also contributed significant levels of Treg-specific chemokines suggesting a key role for both immune and non-immune components in shaping the TME into an immunosuppressive environment during tumourigenesis (Fig. 5f)[43].

### PDAC tumour macrophage and CD8 T Cell infiltration have distinct association with outcome

To establish the presence and clinical implications of the AE and ME groups, we performed exploratory analysis of a cohort of multiplexed IHC data collected from primary, treatment-naïve samples from 486 patients in the APACT clinical trial20 (see 'Methods'), and quantified CD8+ T cell and CD163+ M2 macrophage densities. We found that CD8+ T cell infiltration correlates with CD163+ macrophage infiltration across all patients (Fig. 6a). We subdivided patients into CD8+ high vs. CD8+ low, and CD163+ high vs. CD163+ low groups, which highlighted a subset of patients with relatively high CD8+ T cell infiltration and low CD163+ macrophage infiltration (equivalent to the AE patient group), and a subset with relatively low CD8+ T cell infiltration and high CD163+ macrophage infiltration (equivalent to the ME patient group). An example of the distinctive staining of samples from each of these

two groups is shown in Fig. 6b. When we performed survival analysis across all four patient groups (Fig. 6c), we found that the ME-equivalent group had the shortest overall survival (OS < 30 months) while the AE-equivalent group had the longest OS (OS > 43 months, p value = 0.0016). Notably, we did not observe significantly distinct survival rates between the patient group with high CD8+ and CD163+ infiltration, compared to the group with low CD8+ and CD163+ infiltration, suggesting that interactions between the two cell types—i.e., the ratio of CD8+ T cells to immunosuppressive macrophages—plays a significant role in patient responses to adjuvant chemotherapy.

## Discussion

Our work sheds light on the potential mechanisms that might underlie the observed differences between myeloid-enriched and adaptive-enriched PDAC tumours. Combining scRNA-seq, CITE-seq and TCR and BCR repertoire analysis of matched blood and tumour samples allowed, the identification of different patient groups with distinct immune cell infiltration, selection, differentiation and response mechanisms within the TME, providing a rational way for the selection and design of novel immunotherapeutic interventions for PDAC patients. Moreover, we validated our primary findings in a large, independent clinical cohort using multiplexed IHC. To this end, we developed several bioinformatics tools, including (1) *SVMCellTransfer* which allows for efficient and effective annotation of published scRNA-seq datasets based on a reference high-confidence dataset, (2) *scClonetoire* which quantifies the intra- and inter-subset clonality and other repertoire metrics run on single cell multi-omics repertoire data, (3) *scRepTransition* which quantifies the clonal overlap between B or T cell subsets within a sample or between samples. Importantly, *scClonetoire* and *scRepTransition* account for sampling depth differences between samples thus ensuring the ability for statistical comparisons between samples. This comprehensive analysis of the immune landscape within treatment-naive PDAC patients provides a valuable scMulti-omics dataset with high-confidence annotations and important insights into the TME.

Through our multi-omics analyses, we show that dominant immune mechanisms within AE patients, characterised by a low infiltration of myeloid cells and increased proportion of lymphocytes, include dysfunctional GC (or TLS) responses, lower isotype switching, higher occurrence of IgM+ B cells, and lower generation of plasma cells. The predominance of intra-tumoural memory B cells and the elevated cell-cell interaction signals between B and T cells suggests an antibody-independent role of B cells, such as antigen presentation. Indeed, the predominant contributors of professional APCs in AE patients were B cells, highlighting a potential role for B cells in PDAC TME in shaping T cell activation[44,45]. However, the poor class switching and SHM in AE patients is indicative that some factors that are needed for TLS formation and cell recruitment could be defective, hampering the full development of a GC-like response. T cells, on the other hand,

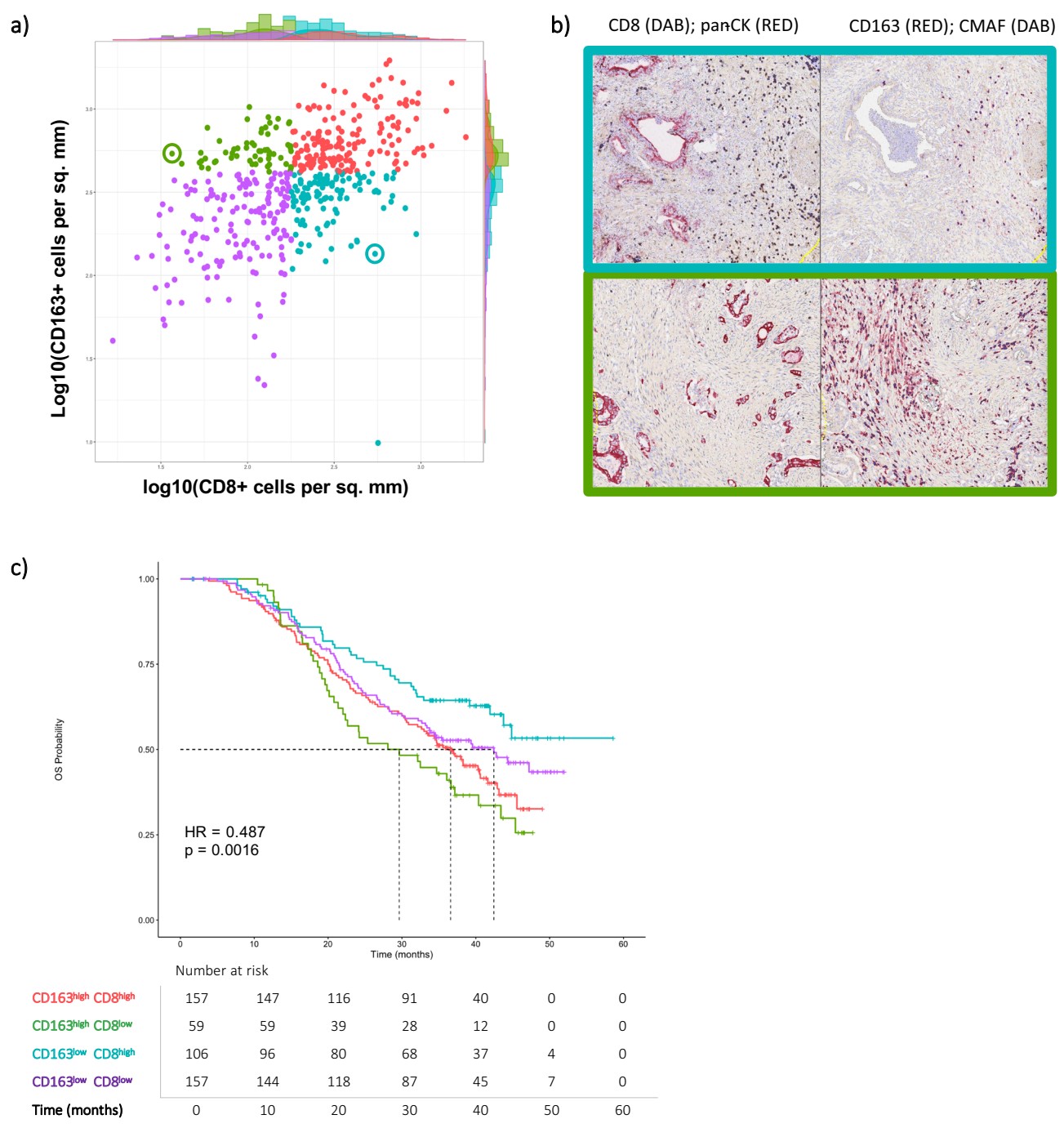

**Fig. 6 | Tumours that are CD163+ enriched and CD8+ depleted exhibit shortest overall survival. a** Densities of CD163+ and CD8+ cells as measured via IHC are correlated across patients in the APACT trial. Circled examples are shown in (**b**) Top: IHC images from an example CD8+ high, CD163+ low sample. Bottom: IHC images from an example CD8+ low, CD163+ high samples. Left are images with CD8+ cells stained in DAB (brown) and pan-CK stained in FastRed. Right are images of aligned sections with CD163+ cells stained in FastRed, and CMAF+ cells stained in DAB (brown). **c** Kaplan–Meier curves shown for the four patient groups defined in (**a**).

showed clearly increased clonality and higher levels of cytotoxic T cells, yet their ability to control tumour was limited, possibly due to poor infiltration or retention outside the tumour core due to high stromal expression of the CXCR4 ligand, *CXCL12* or/and the adoption of senescent phenotype. This exclusion of cells is further supported by our additional data where we see the strongest survival benefit with CD8 T cells in close proximity to the tumour.

ME patients have exhibited a poorer survival signature across multiple studies and are characterised by the higher infiltration of myeloid cells. Our clinical data also confirms that elevated levels of CD163+ macrophages with reduced levels of CD8+ infiltration into the

tumours result in worse overall survival. Here we showed that ME tumours have a higher presence of immuno-regulatory moMacs and mregDCs. Indeed, we show in our cohort that myeloid cells can potentially act as coordinators of further immuno-suppressive mechanisms through extensive cell-cell signalling mechanisms that are distinct from AE patients, including the attraction of regulatory T cells into the TME[46]. Treg cells in ME patients are highly expanded from naive T cells, likely driven by the high levels of TGFb in the pancreatic TME[47,48]. Their association with Tfh T cells could be a driver of differentiation of Tfr (T follicular regulatory cells) which further limit the GC B cell responses in those patients[49,50]. Reduced GC B cell

clonal expansion but increased plasma cell fate in ME patients points to direct macrophage-plasma cell cross-talk inducing plasma cell differentiation[23]. GC B cells in ME patients were not as clonal as in AE patients, however, the plasma cell responses are predominantly via *IGHA1* and *IGHA2*. Indeed, the IgA isotype can engage with the inhibitory Fc receptor *FCAR* on myeloid cells, and can mediate inhibitory effects on many immune cell subsets via activation of FcαRI receptors and induction of IL10 production[51]. Previous studies have shown that tumour-associated antibodies may also exert a pro-tumoural effects through inflammation initiation and maintenance and tissue remodelling[52,53].

We further reveal differential B and T cell selection within the tumour between ME and AE patients. Their infiltration into the tumour may also be limited in ME patients due to the lack of upregulation of key chemokine receptors upon entry into the tissue, including *CXCR4* and *CXCR5*, which have been shown to be important for control of B and T cell trafficking into tissues and play central roles in orchestrating the adaptive cell functions[54]. Indeed, CXCR4 upregulation is known to be driven by factors including hypoxia (HIF1A and VEGF)[55], where the pancreas is a significantly more hypoxic environment than the blood[56]. However, previous studies have shown that extent of hypoxic areas within the tumour correlates with worse survival of PDAC patients[56]. Alternatively, strong antigen signalling has been shown to downregulate CXCR4 in T cells in melanoma[57]. However, stronger antigen signalling in the ME patients is not well supported due to the higher Treg levels, lower T cell infiltration levels, and lower levels of B-T cell interaction signals. This lays the foundation for future studies determining the key molecular factors influencing lymphocyte infiltration and egress.

Overall, we can highlight numerous potential mechanisms that might underlie the observed differences between ME and AE patients and identify two potential major themes for immune intervention within PDAC patients. In ME patients, targeting the inhibitory myeloid compartment alongside specific targeting of tumour infiltrating Treg cells may have the ability to alleviate some of the suppressive mechanisms. For example, the CCR4 and CCR8 pathways are highly active in Treg cells and appear to play a crucial role within the TME and their interactions with myeloid cells. These highly activated Treg populations in ME patients that have multiple checkpoints/inducible co-stimulators activated including targets of immunotherapy including TIGIT/PVR, ICOS/ICOSLG and CTLA4/CD80/CD86[58]. It is important to note that the interplay within the TME between the different populations is complex and redundant effects could be at play, as previous reports have suggested that depleting Treg cells or fibroblasts can both result in worsening of the disease through the conversion by TGFb of a pathogenic myeloid population even as CD8 T cell responses can be improved[46,59–61]. Here, we identify chemokine pathways that potentially lead to a suppressive TME by both immune and non-immune cells, including targets of immunotherapy CCR8, CCR4 and CXCR4 and their ligands, as well as checkpoints TIGIT/PVR and SIRPA/CD47[62]. It is therefore critical to start considering those interventions in combination.

AE patient tumours contain diverse lymphocyte subsets, and their activation status suggests that sufficient neoantigens are presented to them. However, the immune-incompetent TME is potentially preventing proper anti-tumour immune responses as can be seen with high levels of *CXCR4* on the B cell compartment, potentially restricting their access to the tumour core via retention outside. Indeed, higher number and a specific locations of B cells quality in TME, maturity of TLSs, and neoantigen have been shown in PDAC long-term survivors[16]. These data suggest that patients with higher adaptive cell infiltration may benefit most from boosting the immune response against abortive or dysfunctional TLSs, which may potentially be achieved by cancer vaccines[63], targeting T cell senescence and/or targeting chemokines. Conversely, patients with higher myeloid cell infiltration may

benefit most from selective targeting of Treg functions, such as with anti-CCR8[35,64,65] and plasma cells. Our comprehensive analysis generates a myriad of hypothesis to be mechanistically investigate using advanced in vitro and in vivo models as well as in immunotherapy-based clinical trials.

This study lays the foundation for understanding why immunotherapy has not been successful in PDAC so far and provides an avenue for designing novel therapeutic targets based on a complete understanding of patient intra-tumoural immune heterogeneity. Intra-tumoural heterogeneity and sampling remain a limitation for quantifying and comparing tumour infiltrating immune cells and single cell studies can only capture the immune cells from a portion of a heterogeneous tumour and therefore may not represent the whole PDAC tumour within each patient. However, this work shows that we can capture a wide diversity of immune cells, where differences between the captured regions from individual tumours can be investigated at unprecedented detail. We demonstrate the need for trials to assess changes in immune infiltration over time between different therapies to build a spatio-temporal understanding of the tumour-immune cross-talk dynamics. Overall, this framework, which combined multi-modal data, integrated knowledge-based, unsupervised micro-structural annotations, and computational tools, has the power to drive niche discovery and can be applied to other tissues in health and disease, such as in cancers with similar AE and ME differential prognostic signatures including glioblastoma[66], breast[67], prostate[68], non-small cell lung cancer, melanoma[69], bladder cancer[70,71].

## Methods

### Ethics approval and consent to participate
Informed consent was obtained for all patients. The study was in strict compliance with all institutional ethical regulations.

### Sample access and preparation for scRNA-seq
Patients who underwent a curative resection for pancreatic ductal adenocarcinoma were consented for this study. Eight patients were recruited from Oxford under the Oxford Radcliffe biobank (09/H0606/5 + 5, project: 19/A177, REC: NRES Committee South Central - Oxford C). Four patients were recruited from Aachen medical centre under RWTH Aachen biobank project: EK360/19. Informed consent was obtained for all patients. The study was in strict compliance with all institutional ethical regulations. All tumour samples were surgically resected primary pancreatic ductal adenocarcinomas. All tumours were subjected to pathological re-review and histological confirmation by two expert PDAC pathologists before analysis. A supplement providing individual clinical information is provided as Supplementary Data 2.

The methods for sample collection, PBMC isolation and tissue digestion were previously designed in our manuscript Sivakumar et al.[7]. Briefly, pancreatic tissue was collected shortly after resection. Sample was initially mechanically disrupted using a scalpel into small pieces. The pieces were put into a 15 mL conical tube, with 9 mL of complete RMPI (10% FBS, 1% Pen/Strep and 1 mM Glutamine) and 1 mL of 10x hyaluronidase/collagenase solution (StemCell, 07912, Vancouver, BC, Canada). A first round of digestion was done at 37 °C for 30 min in a pre-warmed shaker. The supernatant was collected without disrupting the tissue and a fresh digestion media was added (10 mL complete RPMI containing 200 U of collagenase IV (Lorne Laboratories, LS004194, Danhill, Berkshire, UK), 100 mL/mL of DNAaseI (Sigma, DN25, Gillingham, Dorset, UK) and 0.5 U of universal nuclease (Pierce, 88702, Waltham, MA, USA) for an additional 30 min of digestion as before. The supernatant was combined with the one from the first digestion step and the remaining tumour pieces were squeezed through a 100 mm tissue strainer with a further 10 mL of complete RPMI. The supernatants from all digestion steps were combined and centrifuged for 10 min at 300 × *g*. Any residual red blood

cells were removed with ACK solution. The cells were stained using a viability dye and anti-CD45 antibody diluted at 1:200 (Clone 2D1 PECy7, 368532, Biolegend) and sorted on an LSR Aria. Following the sorting the cells were kept overnight in the fridge and stained with a CITEseq TotalSeqC antibody panel. For the full list of CITEseq antibodies see Supplementary Data 9.

### scMulti-omics sequencing and pre-processing
scRNAseq transcriptome processing was performed using the Chromium 10x system involving GEM generation, post GEM-generation clean-up, cDNA amplification and DNA quantification. The library was sequenced using the Illumina NovaSeq platform. Chromium Single Cell Immune Profiling Reagent Kits v1.1 solution was used to deliver a scalable microfluidic platform for digital CITEseq (Cell Surface Protein), GEX, VDJ TCR and VDJ BCR by profiling 500–10,000 individual cells per sample. Libraries were generated and sequenced from the cDNAs and 10x Barcodes were used to associate individual reads back to the individual partitions.

The analysis pipeline applied to process Chromium single-cell data to align reads and generate feature-barcode matrices was performed as previously described[72]. Briefly, gene expression FASTQ files were processed using Cellranger count (v3.1.0) to perform alignment, filtering, barcode counting, and UMI counting, using 10X Genomics' GRCh38 v3.0.0 reference for Gene Expression analysis and IMGT's reference for VDJ TCR and BCR analysis. It uses the Chromium cellular barcodes to generate feature-barcode matrices, determine clusters, and perform gene expression analysis.

### Filtering, doublet detection and batch correction of the PancrImmune dataset
For each sample, cells with fewer than 500 transcripts or 500 genes were filtered out. Normalisation and scaling was done using the standard Seurat pipeline. Principal component analysis (PCA) was performed on 5000 highly variable genes (HVGs) to compute 50 principal components, then *Harmony* was performed (reference) for batch correction, UMAP for dimensionality reduction, and the Louvain algorithm was used for clustering. These clusters were then annotated broadly into B cell, T cell or myeloid clusters based on mapping of >10% BCR+ droplets and elevated CD19 expression, >10% TCR+ droplets and elevated CD3 expression, <10% BCR/TCR+ droplets, respectively.

Doublet identification and removal was performed using both DoubletFinder[73] and MLtiplet[74]. Each cell type was subsetted into individual objects, and re-clustering within these objects was performed excluding genes which were likely to be influenced by experimental rather than biological factors[75]. These include genes encoding for TCR variable chain, ribosomal proteins, heat shock proteins, mitochondrial proteins, cell cycle proteins, HLA, and noise-related genes (MALAT1, JCHAIN, XIST). For the B and T cell objects, immunoglobulin variable, TCR variable and isotype genes were also excluded.

### Cell type annotations
**T/NK cell annotations of the PancrImmune data.** The re-dimensionality reduced T cell object resulted in 100 clusters generated by k-means. Where ADT-seq data was available this was used in preference to RNA for annotation. T cell clusters were defined as those with mean proportion TCR expression >0.3, with innate clusters being those with mean proportion TCR < 0.3. Individual cells in innate clusters which expressed TRA or TRB sequences were labelled as NK-like T cells.

The innate cells were re-clustered without the T cells to generate 10 clusters, and were labelled by gene expression, ILC1 (*TBX21, IFNG, CCL3*), ILC3 (*RORC, AHR, IL23R IL1R1*), gdT (*TRDC*), NK (*EOMES, GZMA, GNLY, KLRC1*) based on de Andrade et al.[76]. CD56 bright (immature) NK

cells were labelled based on ADT-seq CD56 expression. The remaining NK clusters were labelled based on gene expression patterns to give phenotypic descriptions. NK transitional cells have greater expression of cytokines, chemokines and their receptors (*XCL1, XCL2, CXCR4*), NK mature cells have greater expression of cytotoxic genes (*GZMA, GZMB, PRF1*), NK terminal cells have greater expression of adaptive genes (*PRDM1, ZEB2*).

CD4 and CD8 clusters were defined by ADT-seq expression. As has been well documented in T cell single cell papers, there were clusters with overlapping CD4 and CD8 expression. Cells in overlapping clusters were reassigned at the single cell level if either CD4 or CD8 expression was higher. Memory phenotypes were label based on CD45RA, CD45RO, and CD62L expression. Naïve (CD45RA, CD62L), EMRA (CD45RA), EM (CD45RO), CM (CD45RO, CD62L). Further phenotypic labels were based on RNA expression. Exhausted (4 or more of the following: *HAVCR2, PDCD1, TOX, LAG3, CTLA4, TIGIT, CD38, ENTPD1*). CD4 cells: Treg (*FOXP3*), senescent (*B3GAT1, KLRG1, CD28-, CD27-*), Tfh (*BCL6, ICOS, CXCR5*), Th17 (*RORC*), Th2 (*GATA3*), Th1 (*TBX21*). Finally, clusters were labelled as activated based on HLA-DR ADT-seq expression.

### B cell annotations of the PancrImmune data
The re-dimensionality reduced B cell object resulted in 34 clusters generated by Louvain clustering, and AddModuleScore was used to identify enriched phenotypes (Supplementary Data 10). Plasma cells were defined as clusters with the percentage of droplets above the 95th percentile BCR nUMIs (percBCR_high) >40% and PC score>0.04, plasmablasts as percBCR_high >15%, naive B cells with >80% unmutated BCRs and >98% IGHD/M BCRs, and memory B cells with mean CD27 expression >0.1. The following cell types were based on AddModuleScores and mean gene expression: B cell memory activated (>0.3 activated score and *CD27* expression >0.1), B cell activated pre-memory (>0.4 activated score and *CD27* expression <0.1), B cell MZ (>0.8 FGR score and CD27 expression >0.1), B cell GZMB+ memory (*GZMB* expression>0.3 and *CD27* expression >0.1), B cell pre-GC (>0.2 GCB_FT or >0.02 preGC score), B cell GC (>0.3 GC score), of which B cell DZ GC (>0.9 DZ GC), B cell LZ GC (>0.3 LZ GC score). Finally, naïve B cells were reassigned at the single cell level if there was >3 SHM, if the isotype was not *IGHD/M*, or if there was detectable *CD27* expression (activated memory) or without *CD27* expression (activated pre-memory).

### Myeloid cell annotations of the PancrImmune data
The re-dimensionality reduced myeloid subsetted object was used to identify enriched phenotypes (Supplementary Data 10). We down-sampled the cells to 2000 UMIs/cells and selected variable genes similarly to the seeding step of the clustering. To focus on biologically relevant gene-to-gene correlation, we calculated a Pearson correlation matrix between genes for each sample. For that purpose expression values were log transformed (log(1 + UMI(gene/cell))) while genes with less than 5 UMIs were excluded. Correlation matrices were averaged following z-transformation. The averaged z matrix was then transformed back to correlation coefficients. We grouped the genes into gene modules by complete linkage hierarchical clustering. Specifically, semi-supervised module analysis by complete linkage hierarchical clustering was carried out on variable, biologically-meaningful, and abundantly expressed genes[77]. For example, curated cell-cycle genes and other lateral programmes (such as HLA− and HIST−) were excluded from module analysis. Subsequently, myeloid cells were assigned annotations at two levels of granularity based on prior knowledge of marker genes and modules, spanning PDAC and other cancer datasets.

### Annotation of published datasets using SVMCellTransfer
The raw gene-count matrices from Steele et al. and Peng et al. were downloaded from refs. 10,12, filtered using the same parameters, as

above, and merged. The B, T and myeloid cells were identified, separated in individual objects, merged and batch corrected with the PancrImmune populations via Harmony, and annotated using the custom-written support vector machine (SVM) cell label transfer method, *SVMCellTransfer* (https://github.com/rbr1/scIsoTyper/). SVMs are well established for handling high-dimensional data and complex feature interactions including capturing non-linear relationships[78] which are the challenges frequently encountered with single cell annotations. Our approach offers several advantages over existing annotation models, firstly, most existing methods do not offer the users to assign their own cell reference for annotating dataset of unknown cell types (scAnnotate[79], singleR[80], CellAnn[81], ScType[82], scMayoMap[83], Azimuth[84], CellTypist[26]). Here, we use our PancrImmune data as a reference to annotate the Peng and Steele datasets. This limits the accuracy of the annotations as the references provided are either (1) not tissue-/disease-specific and therefore do not reflect the true nature of cells within the query dataset, (2) do not allow for expert knowledge of a particular tissue/disease to be applied, (3) will not reflect rare cell populations, such as those associated with tissue adaptations or particular functionality that is not reflected in the reference, and (4) do not allow for the user to define how well the reference and query datasets are integrated for label transfer. Some of these methods are also restricted to web server interfaces (CellAnn[81] and Azimuth[84]), which is not appropriate for large datasets. In contrast, SVMCellTransfer allows the user to use the annotation of any other dataset, such as those reflecting the same tissue and disease as performed here and annotated using domain experts, then to perform their own batch correction and integration methods across reference and query datasets to ensure accurate overlay of cells between datasets. We apply subsampling methods for unbiased cell type annotation which ensures that cell annotation is not based on probability of higher number of a cell type but based on gene expression, and provide a probability score for each annotation. Finally, current web-server based methods are not scalable for large datasets. Hence, implementation of this computational method is a useful resource for performing high-confidence cellular annotations at scale for the single cell analysis community.

To validate the performance of SVMCellTransfer, we subsampled high-confidence T and NK cells from the Peng and Steele datasets (confirmed through manual checking of key T and NK genes), and applied both the SVMCellTransfer (using T and NK cell PancrImmune reference), and Azimuth annotation (using either pancreas or PBMC references) (Supplemental Data 1 Fig. 6). Using SVMCellTransfer, we identified the broad set of T and NK populations as expected (top-left), whereas the Azimuth (pancreas reference) identified some T/NK cells as non-immune (top-right). The Azimuth (PBMC reference) identified most cells as T/NK cells, however, mislabels some as B cells or myeloid cells (bottom). Therefore, we recommend using SVMCellTransfer with the appropriate reference, especially when multi-modal data such as VDJ-seq and CITE-seq has been used along with cell-type expertise has been used.

The non-immune cells from the Peng et al. and Steele et al. datasets were merged and batch-corrected, then broad cellular annotations were performed using published cell-type markers (Supplemental Data 1 Figs. 7 and 8).

### BCR-seq/TCR-seq analysis
A pipeline, scIsoTyper, was written to assign most probable BCR IGH and IGK/L chains per droplet (based on nUMIs) and most probable TCR TRA and TRB chains per droplet (based on nUMIs). Briefly, scIsoTyper performs the following steps: (1) batches TCR and BCR data for IMGT submission; (2) parses the IMGT annotation results, for the quantification of SHM (using the IGHV identity output of IMGT), V/J gene usages, and CDR3 sequence identity; (3) identifies the highest expressed light chain and heavy chain BCR sequences per droplet, and

the highest expressed alpha chain and beta chain TCR sequences per droplet; (3) clonality analysis between all samples from the same individual (allowing for assessment between blood and tumour). This is performed using a single-cell extension of established VDJ network construction software from Bashford-Rogers et al.[25] as part of scIsoTyper. A clone refers to a group of clonally related B cells, each containing BCRs with identical CDR3 regions and IGHV gene use, or differing by single point mutations, such as through SHM. Each cluster is assumed to arise from the same pre-B cell. Clones were defined as a combination of (a) group of B cells expressing BCRs that are connected by SHM (i.e. within a cluster in a BCR network, where edges are generated between BCRs differing by one mutation), as previously described in ref. 25, and (b) group of B cells expressing BCRs with identical CDR3 region DNA sequences and identical IGHV and IGHJ family identities. This definition allows for capturing clonal B cells expressing BCRs that have undergone SHM both within the CDR3 region and outside. (4) Finally, all this information is brought together in a meta-data format to include as part of the Seurat analysis, and provides statistics on the number of droplets with heavy and/or light chains and TCR alpha and/or beta chains.

*scClonetoire* was written to quantify the intra- and inter-subset clonality and other repertoire metrics run on single cell multi-omics repertoire data. Intra-subset clonality measures the number of B, CD4 or CD8 T cell clones with 2 or more cells within each cell subset. This accounts for sampling depth differences between samples through generating a mean across 1000 subsamples a set depth of each sample ($n = 5$ cells). Inter-subset clonality measures the percentage of B, CD4 or CD8 T cells of each cell type as members of clones 3 or more cells across all populations. This accounts for sampling depth differences between samples through generating a mean across 1000 subsamples a set depth of each sample ($n = 50$ cells). These sampling depths were chosen to ensure values were captured across as many immune cell subsets as possible, even when the cell type was rare, whilst still ensuring representation across the sample.

The quantification of clonal overlap between B, CD4 or CD8 T cell subsets within a sample or between samples was performed using a pipeline called *scRepTransition*. For the clonal overlap between B, CD4 or CD8 T cell subsets, the absolute number of B or T cells within the same with different cellular annotations was quantified. For all samples in which 1 or more clonal overlaps between cellular annotations was observed, these were normalised to sum to 1. The relative proportions were statistically compared between patient groups via two-sided MANOVA.

Viral TCR detection was performed using VDJdb[85], McPAS[86] and TCRdb[87] as reference datasets.

### Cell-cell interaction analysis
Existing methods[36,37] for analysing cell-cell interactions from scRNA-seq data suffer from (1) the non-specific nature of the reference to immune cell signalling, with the abundance of false-positives and unrelated changes being observed, (2) the lack of normalisation between different samples, and (3) the inability to consider cell-intrinsic differences only, rather than considering cell number. Here, we addressed all three aspects through a cell-cell communication tool. Firstly, reference receptor-ligand database was immune-cell-specific, generated through intercepting the human ligand-receptor database from Fantom (https://fantom.gsc.riken.jp/) with the genes that were captured in the PancrImmune scMulti-omics dataset (Supplementary Data 7). Secondly, the signalling strengths between each pair of cell subtypes for each receptor-ligand pair was calculated by multiplying the percentage of cells per cell subtype expressing each respective gene for each patient. This was calculated for each cell subtype with ≥3 cells. Thus, the interaction strengths are independent of the total proportions of each cell type within the tumour or blood, and are normalised between each sample. This was plotted using *igraph* in R,

and two-sided MANOVA was used to determine statistical differences between patient groups. The number of inbound and outbound links between cell subtypes was the counts of all corresponding non-zero receptor-ligand signalling strengths (Fig. 5b). This was computed using *igraph* in R. Ranked interaction strengths per cell type were extracted per receptor-ligand pair for the Treg-specific analysis (Fig. 5c).

## APC analysis

The pAPC score for each cell (which quantifies the feature expression programme for MHC II and accessory pathway molecules) was calculated using the AddModuleScore using the pAPC pathway genes (Supplementary Data 10). The distributions of scores from DCs (known pAPCs) and CD8 T cells (known non-pAPCs) was used to define a threshold above which we defined cells as being pAPCs using a logistic regression classifier via fitting a glm in R. The statistics between the proportions of tumour pAPC cells between patient groups was performed using two-sided MANOVA.

## Differential gene expression analysis and pathway analysis

Pseudobulk differential gene expression methods were employed using the edgeR[88] package for analysis of aggregated read counts per cell type per patient. This was chosen to reduce the false positive detection rates, reduce biases between patient samples, and address the problem of zero inflated scRNAseq expression data. Briefly, for cells of a given type, we first aggregated reads across cells within each patient. The likelihood ratio test as well as the quasi-likelihood *F*-test approach (edgeR-QLF). For limma, we compared two modes: limma-trend, which incorporates the mean-variance trend into the empirical Bayes procedure at the gene level, and voom (limma-voom), which incorporates the mean-variance trend by assigning a weight to each individual observation[89]. Log-transformed counts per million values computed by edgeR were provided as input to limma-trend. Differentially expressed genes were defined as adjusted *p* values < 0.05.

## Survival analysis

Data from the PAAD TCGA (https://portal.gdc.cancer.gov/) was downloaded and normalised. Patients that were not pathologically PDAC including samples with <1% neoplastic cellularity, neuroendocrine, IPMN and acinar cell carcinoma were excluded based on sample annotations (http://api.gdc.cancer.gov/data/1a7d7be8-675d-4e60-a105-19d4121bdebf). R packages survival and survminer were employed. The Kaplan–Meier (KM) curve was plotted using *survfit* in R to observe survival probabilities over time between patient groups (high versus low IGHM expression). Surv_cutpoint() and surv_categorize() was used to determine an optimal cutpoint using maximally selected rank statistics for *IGHM* expression. The cox regression model was used to estimate and compare hazard ratios between IGHM high and low groups.

## Cell composition deconvolution

Deconvolution between the PancrImmune and TCGA datasets was carried out using BayesPrism[90], a Bayesian method to infer cell type fraction. The intersection of genes present in both datasets was identified, and the raw untransformed count data were used. To assign cell type labels, cell types from annotated single-cell data were used. Substantial heterogeneity was accounted for by creating cell state labels through sub-clustering of cell types within each patient. A threshold of 50 cells per cell state was applied to ensure a sufficient number of cells for accurate sub-clustering. Genes related to ribosomes, mitochondria, chromosome X, and chromosome Y were filtered out from the analysis, as their presence could introduce bias. When running prism object, count matrix was used as input type and key was set to NULL since there were no malignant cells present in the PancrImmune dataset, as recommended by the authors. All other parameters were left at their default values. The mean cell expression

was then obtained from get.theta() function. Subsequently, downstream analysis included PCA to divide TCGA cohort as myeloid high, adaptive enriched followed by plotting the proportions.

## APACT Trial IHC analysis

APACT Trial is a phase III, multi-centre, international, open-label, randomised trial of adjuvant nab-paclitaxel plus gemcitabine (nab-P/G) vs. gemcitabine (G) for surgically resected pancreatic adenocarcinoma (EudraCT 2013-003398-91; ClinicalTrials.gov identifier: NCT0196443020). The Protocol (online only) and informed consent forms were approved by each study site's independent ethics committee or institutional review board before study initiation. This study was conducted in accordance with Good Clinical Practice, as denoted in the International Council for Harmonisation E6 requirements, and with the ethical principles outlined in the Declaration of Helsinki. Written informed consent was obtained before any study-related procedure. All primary resected pancreatic cancer tissues used for IHC were formalin-fixed, paraffin embedded (FFPE). Five serial sections from each block were cut at 4 microns and stained with 5 sets of dual-colour IHC assays in the order indicated in Supplementary Data 11. The dual IHC assays were performed on Leica Bond III automated stainer (Leica Microsystems, Buffalo Grove, IL, USA) by sequentially adding the primary antibodies (Supplementary Data 11) and respective Leica detection systems. The first primary antibody (listed on the top in the dual assay) was added followed by the Bond Polymer Refine Detection (DAB; DS9800; Leica Microsystems Inc.). A heating step was introduced at 90 °C for 5 min in 1x Bond Wash Solution (AR9590) to eliminate any possible cross-reactivity. Dako Protein Block (Catalogue No. X0909) was used for 5 min immediately before applying the second primary antibody in the dual assay, followed by Bond Polymer Refine Red Detection system (DS9390; Leica Microsystems Inc.). After the dual IHC procedure, slides were counterstained with hematoxylin for 2 min on Bond III autostainer, rinsed in tap water, and baked in a 60 °C oven for 20 min or until completely dry and coverslipped by the Tissue-Tek Film Automated Coverslipper (Sakura Finetek USA, Torrance, CA, USA).

## APACT Trial image analysis

The 5 stained serial sections were scanned using Aperio AT2 scanner (Leica Biosystems, Buffalo Grove, IL, USA) at ×20 magnification. Digital images were imported to HALO software (Indica Labs, Albuquerque, NM, USA) for image analysis. Images from five serial sections were registered using HALO's elastic registration algorithm to spatially align all images. A pathologist manually annotated the PanCK image to delineate the whole tumour region. Artifacts such as folds and out-of-focus areas were excluded. A random forest machine learning tissue classifier was developed using the central cytokeratin-stained slide, to further delineate tumour vs. non-tumour regions, and this tumour/non-tumour mask was applied to the five registered serial sections. For each biomarker pair, the MultiplexIHC algorithm was optimised to segment the nuclei based on the hematoxylin stain, and to classify each nuclei as positive or negative for each biomarker (DAB, FastRed) based on optical density threshold in the appropriate cellular compartment (nucleus, cytoplasm, membrane). Analysis outputs used for this manuscript included densities of marker-positive cells (per mm2) in tumour and non-tumour regions. All algorithm optimisation and QC was performed and verified with support from a pathologist. Images were excluded from downstream analysis if any point in the analysis pipeline was judged to have failed QC. Out of 608 samples collected and processed for IHC, 486 passed QC for both CD8 and CD163 quantitation.

## APACT Trial patient stratification and survival analysis

Patients were divided into CD8+ high vs. CD8+ low and CD163+ high vs. CD163+ low groups using optimal cut point selection via maximally

selected rank statistics, with each cut point constrained to be between 45% and 55% percentiles (i.e., within 10% of the median). These same cut-points are used to subdivide the patients in Fig. 6a. Hazard ratios and associated p-values are reported from continuous Cox proportional hazards ratio regression comparing the described patient subsets.

## Reporting summary

Further information on research design is available in the Nature Portfolio Reporting Summary linked to this article.

## Data availability

Data are available in study SCP2405 in the Single Cell Portal (https://singlecell.broadinstitute.org) and in the VDJserver (https://vdjserver.org/community/7425252850300752366-242ac117-0001-012). Raw single cell data are available in the EGA database under accession code EGAD50000001008. All other data are available in the article and its Supplementary files or from the corresponding author upon request. Source data are provided with this paper.

## Code availability

All code is available via https://github.com/rbr1/scIsoTyper/, https://github.com/rbr1/PancrImmune_PDAC_paper, and https://github.com/sakinaamin/BayesPrism.

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

## Acknowledgements

Firstly, we would like to thank the patients and clinicians who contributed to this study, and to the APACT clinical trial. S.S. was funded on an Oxford-BMS Fellowship with additional funding for the single cells sequencing from BMS and the CRUK Oxford Centre. M.L.D. and J.W. were supported by the Kennedy Trust for Rheumatology Research Cell Dynamics Platform 20 21 17. R.B.-R. and F.T. were supported by the Wellcome Trust, University of Oxford and Oxford Cancer Centre. B.S. was supported by the Association of British Neurologists via the Patrick Berthoud Charitable Trust. M.R.M. and B.F. are supported by the NIHR Biomedical Research Centre at Oxford. S.H. was supported by the NIH NCI transition fellowship K00 CA223043. A.J. was funded by a Clarendon scholarship and S.A. was funded by Clarendon in partnership with St John's College. We thank Theodosios Kyriakou and Angela Lee for overseeing the sample submission at the Wellcome Centre for Human Genetics in Oxford. This work was supported by BMS. The research was supported by the Wellcome Trust Core Award Grant Number 203141/Z/16/Z with additional support from the NIHR Oxford BRC. The views expressed are those of the author(s) and not necessarily those of the NHS, the NIHR or the Department of Health. We also thank Gerard Hoeltzel for comments on the typos and grammar in the manuscript and Shihong Wu for input into the figures.

## Author contributions

S.S. and E.A.-S., M.L.D., and R.B.-R. conceived and designed the analysis. S.S., L.H., F.T., H.S., S.R., I.N., J.W., A.F., G.W., U.N., P.C., L.S., T.E., M.W., S.C., M. Middleton, R.B.-R., M.W., and S.C. collected the data. S.S., A.J., E.A.-B., M.N.L., P.K.S., S.A., S.H., A.M., B.S., S.W., N.M.A., S.R., I.N., B.F., R.B.-R., D.J.R., and T.L. contributed data or analysis tools. A.J., E.A.-B., M.N.L., P.K.S., S.A., S.H., A.M., S.R., I.N., P.V.S., A.R., D.P., M. Merad, M.L.D., R.B.-R., D.J.R., and G.H. performed the analyses. All authors contributed intellectual input/interpretation. S.S., A.J., E.A.-B., M.L.D., E.A.-S., and R.B.-R wrote the paper with input from all other authors. P.K.S., M.N.L., S.A., and D.J.R. contributed equally. M.L.D., E.A.-S., and R.B.-R. jointly supervised this work.

## Competing interests

S.S. held a personal fellowship from BMS during this study with a grant provided to conduct experiments and also has research funding from Alchemab, has received speaker fees and travel grants from Astrazeneca, Novartis and Servier; and also conducts trials with Astrazeneca, Novartis, Roche, Genentech and BioNTech. E.A.B. is a contributor to intellectual property licensed by Oxford University Innovation to AstraZeneca and held a research award from Guts UK/Dr Falk during this project. D.J.R., M.W., S.C., T.L., P.V.S., and A.V.R. are current or former employees and shareholders of BMS. M.R.M. reports grants from GRAIL, Roche, Astrazeneca, BMS, Infinitopes, Immunocore, and study fees from BMS, Pfizer, MSD, Regeneron, BiolineRx, Replimune and Novartis outside of the submitted work. M.L.D. is on the SAB for Adaptimmune and Singula Bio, consults for Molecular Partners, Enara Bio, Labgenius and Astra Zeneca, and undertakes research supported by BMS, Cue Biopharma, Boehringer Ingelheim, Regeneron and Evolveimmune outside the submitted work. The remaining authors declare no competing interests. R.B.-R. is a co-founder of Alchemab Therapeutics Ltd and consultant for Alchemab Therapeutics Ltd, Roche, Enara Bio, UCB and GSK.

## Additional information

Shivan Sivakumar [1,2,3,28] ✉, Ashwin Jainarayanan[2,4,28], Edward Arbe-Barnes[5,6,28], Piyush Kumar Sharma[1,29], Maire Ni Leathlobhair[7,8,29], Sakina Amin [9,29], David J. Reiss [10,29], Lara Heij[11,12], Samarth Hegde[13], Assaf Magen [13], Felicia Tucci [9,14,15], Bo Sun [16], Shihong Wu[9,14], Nithishwer Mouroug Anand [9], Hubert Slawinski[15], Santiago Revale [15], Isar Nassiri [15], Jonathon Webber[2], Gerard D. Hoeltzel[9], Adam E. Frampton [17,18,19,20], Georg Wiltberger[21], Ulf Neumann[21,22], Philip Charlton [1], Laura Spiers[1], Tim Elliott [23], Maria Wang[10], Suzana Couto[24], Thomas Lila[10], Pallavur V. Sivakumar[10], Alexander V. Ratushny[10], Mark R. Middleton [1], Dimitra Peppa[6,25], Benjamin Fairfax [1], Miriam Merad [13], Michael L. Dustin [2,26,30], Enas Abu-Shah [2,27,30] ✉ & Rachael Bashford-Rogers [9,14,15,30] ✉

[1]Department of Oncology, University of Oxford, Oxford OX3 7LF, UK. [2]Kennedy Institute of Rheumatology, Nuffield Department of Orthopaedics, Rheumatology and Musculoskeletal Sciences, University of Oxford, Roosevelt Dr, Headington, Oxford OX3 7FY, UK. [3]Department of Immunology and Immunotherapy, School of Infection, Inflammation and Immunology, College of Medicine and Health, University of Birmingham, Birmingham B15 2TT, UK. [4]Institute of Developmental and Regenerative Medicine (IDRM), Old Road Campus, Old Rd, Roosevelt Dr, Headington, University of Oxford, Oxford OX3 7TY, UK. [5]Oxford University Clinical Academic Graduate School, John Radcliffe Hospital, University of Oxford, Oxford OX3 9DU, UK. [6]UCL Institute of Immunity & Transplantation, The Pears Building, Pond Street, London NW3 2PP, UK. [7]Department of Microbiology, Trinity College, Dublin, Ireland. [8]Oxford Big Data Institute, Old Road Campus, University of Oxford, Oxford OX3 7LF, UK. [9]Department of Biochemistry, South Parks Road, University of Oxford, Oxford OX1 3QU, UK. [10]Bristol-Myers Squibb, Seattle, Seattle, WA, USA. [11]GROW School for Oncology and Developmental Biology, Department of Pathology, Maastricht University Medical Center, Maastricht, The Netherlands. [12]Department of Surgery and Transplantation, University Hospital RWTH Aachen, Aachen, Germany. [13]Precision Immunology Institute, Icahn School of Medicine at Mount Sinai, 1 Gustave L. Levy Pl, New York, NY 10029, USA. [14]Oxford Cancer Centre, Oxford, UK. [15]Wellcome Centre for Human Genetics, University of Oxford, Oxford, UK. [16]Nuffield Department of Clinical Neurosciences, University of Oxford, Oxford OX3 7LD, UK. [17]Minimal Access Therapy Training Unit (MATTU), Leggett Building, University of Surrey, Daphne Jackson Road, Guildford GU2 7WG, UK. [18]Department of Hepato-Pancreato-Biliary (HPB) Surgery, Royal Surrey County Hospital, Egerton Road, Guildford GU2 7XX, UK. [19]Targeted Cancer Therapy Unit, Department of Clinical and Experimental Medicine, Faculty of Health and Medical Science, University of Surrey, Guildford GU2 7WG, UK. [20]Division of Cancer, Department of Surgery and Cancer, Imperial College London, Hammersmith Hospital Campus, London W12 0NN, UK. [21]Department of General, Visceral, and Transplantation Surgery, University Hospital of RWTH Aachen, Aachen, Germany. [22]Department of Surgery Maastricht University Medical Center (MUMC), Maastricht, The Netherlands. [23]Centre for Immuno-oncology, Nuffield Department of Medicine, University of Oxford, Oxford, UK. [24]Neomorph, Inc., 5590 Morehouse Dr, San Diego, CA, USA. [25]Nuffield Department of Medicine, Old Road Campus, University of Oxford, Oxford OX3 7BN, UK. [26]Chinese Academy of Medical Sciences Oxford Institute, Nuffield Department of Medicine, University of Oxford, Oxford OX3 7BN, UK. [27]Sir William Dunn School of Pathology, South Parks Road, University of Oxford, Oxford OX1 3RE, UK. [28]These authors contributed equally (as first authors): Shivan Sivakumar, Ashwin Jainarayanan, Edward Arbe-Barnes. [29]These authors contributed equally (as second authors): Piyush Kumar Sharma, Marie Ni Leathlobhair, Sakina Amin, David J. Reiss. [30]These authors jointly supervised this work: Michael L. Dustin, Enas Abu-Shah, Rachael Bashford-Rogers. ✉e-mail: s.sivakumar@bham.ac.uk; enas.abushah@path.ox.ac.uk; rachael.bashford-rogers@bioch.ox.ac.uk

