## [Peer Review File · Nature Communications]

Distinct immune cell infiltration patterns in pancreatic ductal adenocarcinoma (PDAC) exhibit divergent immune cell selection and immunosuppressive mechanismsREVIEWER COMMENTS

Reviewer #1 (Remarks to the Author):

The manuscript Distinct immune cell infiltration patterns in pancreatic ductal adenocarcinoma (PDAC) exhibit divergent immune cell selection and immunosuppressive mechanisms by Sivakumar et al. is a very nice study showing deep characterization of the immune profile of PDAC tumors. The work demonstrates that PDAC tumors cluster into two major groups, those with a myeloid-enriched (ME) or an adapted-enriched (AE) phenotype. AE tumors present increased proportion of lymphocytes and reduced myeloid cells. The definition of these populations is the result of a multi-omics analysis of a set of 12 untreated PDAC tumors and the integration and re-analysis of two public datasets. Additionally, they have single cell sequencing data from CD45+ cells coming from matched PBMCs of the 12 PDAC patients.

They present robust data embracing different layers of complexity of the immune infiltration. The paper is a detailed bioinformatic study from the multi-omics integration data providing large amount of information that is displayed in comprehensive figures and tables.

The limitation of the study is that the results are restricted to sequencing and analysis of the data but lack experimental validation. Immunohistochemical staining of relevant markers to illustrate the two major groups of ME and AE PDAC phenotypes will be highly relevant. Moreover, such study may help to visualize the potential retention of certain immune cell subsets outside the tumor core as the authors hypothesize.

Minor comments.

Update reference number 1 of Siegel et al. to the 2023 publication.

Reviewer #2 (Remarks to the Author):

The manuscript by Sivakumar et al. describes immune single cell RNA-seq data from 12 PDAC patients, emphasizing on differences between myeloid-enriched and adaptive-enriched tumor microenvironments. It combines additional OMIC analyses (CITE-seq/TCR-seq) and also compares tumor immune infiltrates to those in circulation. The computational analysis is rigorous, and details clonal expansion beyond what is currently available in the literature. It also combines two other datasets previously described expanding the analysis to a large number of immune cells analyzed. The body of work is mostly descriptive, and conclusions are based exclusively on the theoretical characterization. No functional data or validation is provided to support the findings or strengthen the confidence that novel findings are worth pursuing as therapeutic targets or as prognostic markers. The pathways and candidate genes described are backed through prior findings in other reports. The manuscript would be improved through the addition of some validation experiments.

Reviewer #3 (Remarks to the Author):

Overall Summary:

Sivakumar et al. report numerous analyses on their new multi-omic PDAC dataset encompassing 12 patients with matched CD45+ PDAC-infiltrating cells and peripheral blood cells. The authors employ scRNA-seq, CITE-seq, and scTCR/BCRseq, as well as several new computational tools, to identify potential therapeutic targets for treating patients with PDAC. These data are a tremendous resource for the larger PDAC community, and their results have generated multiple intriguing hypotheses. However, the manuscript lacks experimental validation, and the work needs to be clearly identified as a hypothesis-generating study throughout. Additionally, the authors emphasize their new computational tools, but have not sufficiently compared their approaches to other published methods nor described their rationale or methodology in enough detail. Multiple typos and errors in figure and table references and legends made this manuscript difficult to review.

Major points:

1. Although the reviewer is not an expert in using SVM for cell label transfer, the methods (lines 688-695) for SVMCellTransfer lack sufficient detail for a reader to establish even a

general understanding of its implementation. This is essential because it is described as novel in the text and used to annotate data from two disparate datasets (Steele and Peng). Were comparisons to other methodologies completed to gauge performance? The results section (lines 117-134) also lacks clarity, and references to supplemental item 1 should specify panels.

2. It's unclear whether the PBMCs in the Pancrimune dataset are included in the data shown in Figure 1, specifically 1c-1f, or if these are just the PDAC-infiltrating immune cells. This should be clarified to avoid confusion.

3. Figures 1d and 1e are switched in the figure legends.

4. Intratumoral heterogeneity and sampling should be acknowledged and briefly discussed as a limitation when quantifying and comparing tumor infiltrating immune cells. More specifically, we are only capturing the immune cells from a portion of a heterogeneous tumor, and this may not be representative of the patient's PDAC generally.

5. Discussion of Figure 2 should clarify which data are being shown. Although the figure legend for 2a specifies Pancrimune data, this should be obvious in the text and specify whether the PBMC data are also included. Labeling the figure by dataset would also be helpful.

6. Please describe how SHM was quantified in the methods (BCR/TCR-seq analysis, lines 697-721).

7. For the intra and inter-subset clonality metric, how was a B cell clone defined? Did this require an exact V, CDR3, J, and IgH sequence match? This should be specified in the methods.

8. In lines 212-214: "We next assessed the clonality of the B cell subpopulations via two measures: intra-subset clonality which reflects specific cell populations which are actively expanding, and inter-subset clonality to reflect the expansion and differentiation between

subpopulations.” These data are only looking at a single timepoint, and therefore cannot be defined as actively expanding. At most, the authors can state that this clone is generally expanded.

9. Figures 2d and 2e are switched in the figure legend.

10. For the APC analysis methods (lines 735-741), table S7 does not contain a list of pAPC pathway genes. Table S7 is “CITE-seq antibody information used in the PancrImmune dataset”. I believe the appropriate table should be S8.

11. In the “Increased CD8 T cell clonality in AE patients” (line 250), the authors should specify whether the PancrImmune PBMC data are included in these analyses.

12. In the section “T cell clonality between tumour and blood are distinct” (lines 316-332): Although the authors examined known anti-viral TCRs, the presence of expanded clones in the blood could still indicate bystander infiltration of the tumor rather than “tumor expanded” clones re-entering the blood. This is especially important to note, as these patients will have diverse HLAs and T cell responses that may not be fully represented in published tables of known TCR antigen specificity.

13. In the “Distinct T cell clonal fate between AE and ME patient groups” section (lines 302-314), the conclusions regarding activated CD8 T cells moving toward dysfunctional states may be supported by pseudotime or similar single-cell trajectory inference analysis.

14. In the “Immunosurveilling and resident tumour-infiltrating B and T cell clones are phenotypically distinct” section (lines 334-348), it’s important to know the peripheral abundance of the tumor-infiltrating clones they are analyzing. This would be particularly helpful to their hypothesis that certain immune cells may expand intratumorally, although it would be more robust to use bulk TCR/BCRseq of PBMCs to better sample the circulating repertoires (~100000s of immune receptors versus ~1000s of immune receptors with scRNAseq). If this isn’t feasible, reporting the abundance from the single-cell data could still be informative, as we anticipate more abundant clones to be better sampled by scRNA-seq.

15. “Finally, we show that only TIL T cell clones tend to be acutely activated with elevated with CD69 and PD1) whereas the blood counterparts of same clones are not acutely activated (Figure 4d).”, lines 416-419: Data regarding CD69 and PD1 expression in TIL T cell clones does not appear in Figure 4d.

16. More methodologic details about scIsotype are needed (lines 698-701).

17. For the cell-cell interaction analysis, were any comparisons done to other available methods? Either way, more methodologic detail rationalizing this approach is appropriate.

18. The following link does not work: <https://github.com/sakinaamin/BayesPrism> (line 785).

19. The manuscript lacks experimental validation. This is clearly outside the scope of this already large manuscript, but the findings should be clearly discussed as hypothesis-generating and not definitive. Softening the language, particularly in the abstract, introduction, and conclusions, would suffice, as well as clearly stating the need for experimental validation before considering intervention in patients.

Minor points:

1. Some acronyms have not been properly defined prior to their use in the text.

2. Many typos, for example lines 99-100: “Importantly, we developed and applied novel single cell analyses to uncouple the distinct roles and contributions different immune cell populations...” should read “Importantly, we developed and applied novel single cell analyses to uncouple the distinct roles and contributions of different immune cell populations...”

3. Authors should avoid phrases like “the most detailed single-cell analysis” (lines 26-27).

4. In line 417: “...PD1) whereas the blood counterparts of same clones...”, the parentheses should be a comma.

REVIEWER COMMENTS

We thank the reviewers their detailed and insightful analysis of our manuscript. We have addressed the concerns raised and believe this has substantially improved the quality and presentation of the manuscript. Overall, the reviewers were positive about our general approach and appreciate that our data provides additional insight into the immune landscape of PDAC will be a valuable resource for the community. We outline the changes below in a point-by-point basis.

Reviewer #1 (Remarks to the Author):

The manuscript Distinct immune cell infiltration patterns in pancreatic ductal adenocarcinoma (PDAC) exhibit divergent immune cell selection and immunosuppressive mechanisms by Sivakumar et al. is a very nice study showing deep characterization of the immune profile of PDAC tumors. The work demonstrates that PDAC tumors cluster into two major groups, those with a myeloid-enriched (ME) or an adapted-enriched (AE) phenotype. AE tumors present increased proportion of lymphocytes and reduced myeloid cells. The definition of these populations is the result of a multi-omics analysis of a set of 12 untreated PDAC tumors and the integration and re-analysis of two public datasets. Additionally, they have single cell sequencing data from CD45+ cells coming from matched PBMCs of the 12 PDAC patients.

They present robust data embracing different layers of complexity of the immune infiltration. The paper is a detailed bioinformatic study from the multi-omics integration data providing large amount of information that is displayed in comprehensive figures and tables.

The limitation of the study is that the results are restricted to sequencing and analysis of the data but lack experimental validation. Immunohistochemical staining of relevant markers to illustrate the two major groups of ME and AE PDAC phenotypes will be highly relevant. Moreover, such study may help to visualize the potential retention of certain immune cell subsets outside the tumor core as the authors hypothesize.

We thank the referee for their appreciation of our findings.

Action taken:

We have now included data showing the two patient groups (ME and AE) and associated survival data (Fig. 6 and associated text) from the APACT Trial, a phase III, multi-center, international, open-label, randomised trial of adjuvant nab-paclitaxel plus gemcitabine (nab-P/G) vs gemcitabine (G) for surgically resected pancreatic adenocarcinoma (clinical trial number: NCT01964430¹). This supports our finding and others that patients with higher macrophage infiltration and low T cell infiltration have worse overall survival (OS <30 months) compared to high CD8+ T cell infiltration (OS >43 months, p-value=0.0016). We have further associated the expression of key immune cytokines and chemokines contributing to the suppressive TME in those patients and corroborate some of the signatures we saw in our scMulti-omics dataset (from line 482):

“To establish the presence and clinical implications of the AE and ME groups, we analysed a cohort of 468 patients from the APACT clinical trial (see **Methods**). CD8+ T cell infiltration correlates with CD163+ macrophage infiltration across all patients (**Figure 6a**). We subdivided patients into CD8+ high vs CD8+ low, and CD163+ high vs CD163+ low groups. This highlighted a subset of patients with high CD8+ T cell infiltration and low CD163+ macrophage infiltration (equivalent to the AE patient group), and a subset with low CD8+ T cell infiltration and high CD163+ macrophage infiltration (equivalent to the ME patient group), with an example of the staining of samples from each of these two groups shown in

Figure 6b. When we performed survival analysis across all four patient groups (**Figure 6c**), we found that the ME-equivalent group had the shortest overall survival (OS < 30 months) while the AE-equivalent group had the longest OS (OS > 43 months, p-value=0.0016). Furthermore, we performed differential bulk RNA expression analysis between these two CD163+ macrophage and CD8+ T cell HC-defined immune infiltrate categories, and found that the CD163+ high, CD8+ low (ME-equivalent) patient group had higher expression of the CXCL5/CXCR2, IL8/IL8R and CCR8 pathways, all of which contribute to a more suppressive immune microenvironment (**Figure 6d**)."

We have included the additional discussion points as well as materials and methods for this additional analysis.

Minor comments.

Update reference number 1 of Siegel et al. to the 2023 publication.

We have updated this reference to "Siegel, R. L., Miller, K. D., Wagle, N. S. & Jemal, A. Cancer statistics, 2023. CA Cancer J Clin 73, 17-48 (2023). <https://doi.org:10.3322/caac.21763>"

Reviewer #2 (Remarks to the Author):

The manuscript by Sivakumar et al. describes immune single cell RNA-seq data from 12 PDAC patients, emphasizing on differences between myeloid-enriched and adaptive-enriched tumor microenvironments. It combines additional OMIC analyses (CITE-seq/TCR-seq) and also compares tumor immune infiltrates to those in circulation. The computational analysis is rigorous, and details clonal expansion beyond what is currently available in the literature. It also combines two other datasets previously described expanding the analysis to a large number of immune cells analyzed. The body of work is mostly descriptive, and conclusions are based exclusively on the theoretical characterization.

No functional data or validation is provided to support the findings or strengthen the confidence that novel findings are worth pursuing as therapeutic targets or as prognostic markers. The pathways and candidate genes described are backed through prior findings in other reports. The manuscript would be improved through the addition of some validation experiments.

We thank the referee for their appreciation of the added novelty of our manuscript. As a validation and in support of our findings, we have now added clinical association with the two hallmarks of the patient groups we identified (myeloid and adaptive) using data from the AFACT clinical trial. See response to reviewer 1 and figure 6 in the revised manuscript.

Reviewer #3 (Remarks to the Author):

Overall Summary:

Sivakumar et al. report numerous analyses on their new multi-omic PDAC dataset encompassing 12 patients with matched CD45+ PDAC-infiltrating cells and peripheral blood cells. The authors employ scRNA-seq, CITE-seq, and scTCR/BCRseq, as well as several new computational tools, to identify potential therapeutic targets for treating patients with PDAC. These data are a tremendous resource for the larger PDAC community, and their results have generated multiple intriguing hypotheses.

However, the manuscript lacks experimental validation, and the work needs to be clearly identified as a hypothesis-generating study throughout.

We thank the referee for their comments. We agree that we were missing stronger validation of our conclusions. This has now been addressed following the comment from all three reviewers (see answers above). We have now included additional data from a clinical trial associating overall survival to the two patient groups we have identified and have amended the text to highlight which claims can be validated from this additional data and what is an intellectual interpretation and hypothesis-generating findings.

Additionally, the authors emphasize their new computational tools, but have not sufficiently compared their approaches to other published methods nor described their rationale or methodology in enough detail.

We appreciate that additional explanation and clarity about the computational methodology was missing. We have now addressed all of these points as detailed below.

Multiple typos and errors in figure and table references and legends made this manuscript difficult to review.

We thank the referee for pointing out these typos and errors and we have corrected these within figure and table references and legends as detailed below.

Major points:

1. Although the reviewer is not an expert in using SVM for cell label transfer, the methods (lines 688-695) for SVMCellTransfer lack sufficient detail for a reader to establish even a general understanding of its implementation. This is essential because it is described as novel in the text and used to annotate data from two disparate datasets (Steele and Peng). Were comparisons to other methodologies completed to gauge performance? The results section (lines 117-134) also lacks clarity, and references to supplemental item 1 should specify panels.

We thank the reviewer for bring up this valid point and we address all aspects of this:

1. Explanation of the methodology

Additional text (see below) and a supplemental figure has now been added to the manuscript to describe the methodology in more detail.

SVMCellTransfer advantages over established methods:

Annotations of the reference can be performed using cell type expertise and/or integration of multi-modal single cell data, using datasets that reflect well the query dataset

The unbiased cell type annotation ensures that cell annotation is not based on probability of higher number of a cell type but based on gene expression. This can also speed up the subsequent steps when using a large reference.

Allows the user to define how the reference and query datasets are integrated prior to label transfer

Scalable, fast, does not rely on user-defined markers, considers non-linear relationships and does not rely on a web-server interface.

Supplemental item Figure 6a. Schematic of SVMCellTransfer alongside the advantages of this method over established methods (left).

2. Comparison to other methodologies

We have now expanded the text to cover the comparison to other methodologies. Briefly, current methodologies (scAnnotate², singleR³, CellAnn⁴, ScType⁵, scMayoMap⁶, Azimuth⁷, CellTypist⁸) rely on a pre-defined reference that is not user-defined. This limits the accuracy of the annotations as the references provided are either (a) not tissue-/disease-specific and therefore do not reflect the true nature of cells within the query dataset, (b) do not allow for expert knowledge of a particular tissue/disease to be applied, (c) will not reflect rare cell populations, such as those associated with tissue adaptations or particular functionality that is not reflected in the reference, and (d) do not allow for the user to define how well the reference and query datasets are integrated for label transfer. Some of these methods are also restricted to web server interfaces (CellAnn⁴ and Azimuth⁷), which is not appropriate for large datasets. In contrast, SVMCellTransfer allows the user to use the annotation of any other dataset, such as those reflecting the same tissue and disease as performed here and annotated using domain experts, then to perform their own batch correction and integration methods across reference and query datasets to ensure accurate overlay of cells between datasets (a process which is project-specific with no one-size-fits-all method available), and using this to perform the reference annotation using SVMCellTransfer. Finally, SVMCellTransfer can provide a probability score for the annotation, allowing the user to refine any low-confidence annotations where required.

As an example of this, we demonstrate that SVMCellTransfer outperforms Azimuth reference annotation. We subsampled high-confidence T and NK cells from the Peng and Steele datasets (confirmed by manual checking of key T and NK genes, as shown in the supplemental item 1), and applied both the SVMCellTransfer (using T and NK cell PancrImmune reference), and Azimuth annotation (using either pancreas or PBMC references) (**Reviewer Figure 1**). Using SVMCellTransfer, we identify the broad set of T and NK populations as expected, whereas the Azimuth (pancreas reference) identifies some T/NK cells as non-immune (top-right). The Azimuth (PBMC reference) identifies most cells as T/NK cells, however, mislabels some as B cells or myeloid cells (bottom). In contrast, SVMCellTransfer appropriately labels the T and NK cell subsets (top-left).

Reviewer Figure 1. Comparison of reference annotation of T and NK cells using SVMCellTransfer (using T and NK cell PancriImmune reference), and Azimuth annotation (using either pancreas or PBMC references). High-confidence T and NK cells were subsampled from the Peng and Steele datasets (confirmed by manual checking of key T and NK genes), and applied both the SVMCellTransfer (using T and NK cell PancriImmune reference), and Azimuth annotation (using either pancreas or PBMC references). UMAP plots show the distribution of predicted cell types by each method.

We have now edited the methods text to reflect these important aspects (from line 728 in revised manuscript):

“The raw gene-count matrices from Steele et al. and Peng et al. were downloaded from ^{9,10}, filtered using the same parameters, as above, and merged. The B, T and myeloid cells were identified, separated in individual objects, merged and batch corrected with the PancriImmune populations via Harmony, and annotated using the custom-written support vector machine (SVM) cell label transfer method, *SVMCellTransfer* (<https://github.com/rbr1/scIsoTypier/>). SVMs are well established for handling high-dimensional data and complex feature interactions including capturing non-linear relationships¹¹ which are the challenges frequently encountered with single cell annotations. Our approach offers several advantages over existing annotation models, firstly, most existing methods do not offer the users to assign their own cell reference for annotating dataset of unknown cell types (scAnnotate², singleR³, CellAnn⁴, ScType⁵, scMayoMap⁶, Azimuth⁷, CellTypist⁸). Here, we use our PancriImmune data as a reference to annotate the Peng and Steele datasets. This limits the accuracy of the annotations as the references provided are either (a) not tissue-/disease-specific and therefore do not reflect the true nature of cells within the query dataset, (b) do not allow for expert knowledge of a particular tissue/disease to be applied, (c) will not reflect rare cell populations, such as those associated with tissue adaptations or particular functionality that is not reflected in the reference, and (d) do not allow for the user to define how well the reference and query datasets are integrated for label transfer. Some of these methods are also restricted to web server interfaces (CellAnn⁴ and Azimuth⁷), which is not appropriate for large datasets. In contrast, SVMCellTransfer allows the user to use the annotation of any other dataset, such as those reflecting the same tissue and disease as performed here and annotated using domain experts, then to perform their own batch correction and integration methods across reference and query datasets to ensure accurate overlay of cells between datasets. We apply subsampling methods for unbiased cell type annotation which ensures that cell annotation is not based on probability of higher number of a cell type but based on gene expression,

and provide a probability score for each annotation. Finally, current web-server based methods are not scalable for large datasets. Hence, implementation of this novel computational method is a useful resource for performing high-confidence cellular annotations at scale for the single cell analysis community.

To validate the performance of SVMCellTransfer, we subsampled high-confidence T and NK cells from the Peng and Steele datasets (confirmed through manual checking of key T and NK genes), and applied both the SVMCellTransfer (using T and NK cell PancrImmune reference), and Azimuth annotation (using either pancreas or PBMC references) (**Supplemental Item Figure 1.6.**). Using SVMCellTransfer, we identify the broad set of T and NK populations as expected (top-left), whereas the Azimuth (pancreas reference) identifies some T/NK cells as non-immune (top-right). The Azimuth (PBMC reference) identifies most cells as T/NK cells, however, mislabels some as B cells or myeloid cells (bottom). Therefore, we recommend using SVMCellTransfer with the appropriate reference, especially when multi-modal data such as VDJ-seq and CITE-seq has been used along with cell-type expertise has been used.”

2. It’s unclear whether the PBMCs in the PancrImmune dataset are included in the data shown in Figure 1, specifically 1c-1f, or if these are just the PDAC-infiltrating immune cells. This should be clarified to avoid confusion.

We have now changed the figure legend text to clarify the cells shown:

“b) UMAP dimensionality reduction of the intra-tumoural immune cells from the PancrImmune dataset depicting total immune cells (centre), B cells (left), myeloid cells (bottom) and T cells (right).”

3. Figures 1d and 1e are switched in the figure legends.

We have corrected these labels in the figure legend.

4. Intratumoral heterogeneity and sampling should be acknowledged and briefly discussed as a limitation when quantifying and comparing tumor infiltrating immune cells. More specifically, we are only capturing the immune cells from a portion of a heterogeneous tumor, and this may not be representative of the patient’s PDAC generally.

We include a few sentences in the discussion to acknowledge the limitations of this study (from line 595 in revised manuscript):

“Intra-tumoural heterogeneity and sampling remain a limitation for quantifying and comparing tumour infiltrating immune cells and single cell studies can only capture the immune cells from a portion of a heterogeneous tumour and therefore may not represent the whole PDAC tumour within each patient. However, this work shows that we can capture a wide diversity of immune cells, where differences between the captured regions from individual tumours can be investigated at unprecedented detail.”

We believe that the samples obtained for the PancrImmune dataset are well representing the overall tumour because:

- We estimate that between 5-20% of the whole tumour was taken for single cell dissociation and for running on the 10X Chromium. This means that representative immune cells from the 5-20% of the whole tumour were captured.
- We performed scMulti-omics on fresh samples (*i.e.* not after freeze-thawing), and therefore would not suffer from such extreme cellular death/apoptosis affects as those from other datasets. Indeed, freeze-thaw-related cell death usually occurs in a cell-type dependent manner, and thus would bias the resulting cell populations.

Overall, we have made every effort to reduce the effects of artefact and biases in this dataset.

5. Discussion of Figure 2 should clarify which data are being shown. Although the figure legend for 2a specifies PancrImmune data, this should be obvious in the text and specify whether the PBMC data are also included. Labeling the figure by dataset would also be helpful.

We include statements in each figure legend clarifying this:

Figure 1:

- b) UMAP dimensionality reduction of the intra-tumoural immune cells from the PancrImmune dataset depicting total immune cells (centre), B cells (left), myeloid cells (bottom) and T cells (right).
- c) The correlation of (left) B cells and Myeloid cells and (right) myeloid cells and T cells as a proportion of total intra-tumoural immune cells across the three datasets, coloured blue red and yellow for the PancrImmune, Peng and Steele datasets respectively.
- d) The cellular proportions of the broad intra-tumoural immune cell types between myeloid- and adaptive-enriched patients in the PancrImmune dataset.
- e) Principal component analysis (PCA) of the intra-tumoural immune cell proportions per sample, coloured orange for myeloid-enriched (ME) patient samples and grey for adaptive immune cell enriched (AE) patient samples (PancrImmune dataset).
- f) Heatmap of the differences in intra-tumoural cellular proportions between ME and AE patient tumour samples. The colour denotes the proportional skew between ME and AE patients, and * denotes a significant difference between ME and AE patients (p-value <0.05). Statistical tests were performed by MANOVA.

Figure 2: Added:

- All single cell analyses in this figure were performed on the intra-tumoural B cells from PancrImmune dataset.

Figure 3: Added:

- All analyses in this figure were performed on the intra-tumoural B cells from PancrImmune dataset.

Figure 4: Added:

- All analyses in this figure were performed on the PancrImmune dataset using both the blood and tumour samples.

Figure 5: Added:

- All analyses in this figure were performed on the intra-tumoural immune cells from the PancrImmune dataset unless otherwise indicated.

Figure S1: Added:

- All analyses were performed on intra-tumoural cells.

Figure S2: Added:

- All analyses were performed on intra-tumoural cells.

Figure S3: Added:

- All analyses in panels b-d were performed on intra-tumoural cells in the PancrImmune dataset, and panels e-f were performed on intra-tumoural or blood cells in the PancrImmune dataset * denotes p-values<0.05, and tests were performed by MANOVA.

Figure S4: Added:

- All analyses in this figure were performed on the PancrImmune dataset using both the blood and tumour samples.

Figure S5: Added:

- All analyses in this figure were performed on the PancrImmune dataset using both the blood and tumour samples.

Figure S6: Added:

- All analyses were performed on intra-tumoural cells.

6. Please describe how SHM was quantified in the methods (BCR/TCR-seq analysis, lines 697-721).

We include a more detailed description of the quantification of SHM in the methods (from line 775 in revised manuscript) clarifying this:

“...the quantification of SHM (using the IGHV identity output of IMG2)...”

7. For the intra and inter-subset clonality metric, how was a B cell clone defined? Did this require an exact V, CDR3, J, and IgH sequence match? This should be specified in the methods.

We include a more detailed description of the clone definition in the methods (from line 778 in revised manuscript):

“...clonality analysis between all samples from the same individual (allowing for assessment between blood and tumour). This is performed using a single-cell extension of established VDJ network construction software from Bashford-Rogers et al.²⁴ as part of scIsoTyper. A clone refers to a group of clonally related B cells, each containing BCRs with identical CDR3 regions and IGHV gene use, or differing by single point mutations, such as through SHM. Each cluster is assumed to arise from the same pre-B cell. Clones were defined as a combination of (a) group of B cells expressing BCRs that are connected by SHM (i.e. within a cluster in a BCR network, where edges are generated between BCRs differing by one mutation), as previously described in 24, and (b) group of B cells expressing BCRs with identical CDR3 region DNA sequences and identical IGHV and IGHJ family identities. This definition allows for capturing clonal B cells expressing BCRs that have undergone SHM both within the CDR3 region and outside.”

8. In lines 212-214: “We next assessed the clonality of the B cell subpopulations via two measures: intra-subset clonality which reflects specific cell populations which are actively expanding, and inter-subset clonality to reflect the expansion and differentiation between subpopulations.” These data are only looking at a single timepoint, and therefore cannot be defined as actively expanding. At most, the authors can state that this clone is generally expanded.

We thank the referee for this point and have reworded this to reflect the single time point nature of the statistic (from line 212 in revised manuscript):

“We next assessed the clonality of the B cell subpopulations via two measures: intra-subset clonality which reflects specific cell populations which have undergone clonal expansion, and inter-subset clonality to reflect the expansion and differentiation between subpopulations (Figure 2e).”

9. Figures 2d and 2e are switched in the figure legend.

We have now corrected this in the figure legend.

10. For the APC analysis methods (lines 735-741), table S7 does not contain a list of pAPC pathway genes. Table S7 is "CITE-seq antibody information used in the PancrImmune dataset". I believe the appropriate table should be S8.

We have now corrected this in the text (line 832 in revised manuscript), noting that an extra table has been included:

"The pAPC score for each cell (which quantifies the feature expression programme for MHC II and accessory pathway molecules) was calculated using the AddModuleScore using the pAPC pathway genes (Table S9)."

11. In the "Increased CD8 T cell clonality in AE patients" (line 250), the authors should specify whether the Pancrimmune PBMC data are included in these analyses.

We have now updated the figure legend to reflect that the analyses in Figure 3 were performed on the intra-tumoural PancrImmune dataset (see referee point 5). In addition, we have clarified this in the text (line 251 in revised manuscript):

"To understand the drivers behind the increased T cell prevalence in the AE patients, we performed clonality analysis of the T cell populations using the TCR sequencing data from intra-tumoural T cells in the PancrImmune dataset."

12. In the section "T cell clonality between tumour and blood are distinct" (lines 316-332): Although the authors examined known anti-viral TCRs, the presence of expanded clones in the blood could still indicate bystander infiltration of the tumor rather than "tumor expanded" clones re-entering the blood. This is especially important to note, as these patients will have diverse HLAs and T cell responses that may not be fully represented in published tables of known TCR antigen specificity.

We have now updated the text to reflect this valid and important point (line 323 in revised manuscript): "This supports that tumour clones are not detectably enriched for systemic non-tumour-reactive clones. However, we note that as these patients will have diverse HLAs and so T cell responses that may not be fully represented in published databases of known TCR antigen specificity."

13. In the "Distinct T cell clonal fate between AE and ME patient groups" section (lines 302-314), the conclusions regarding activated CD8 T cells moving toward dysfunctional states may be supported by pseudotime or similar single-cell trajectory inference analysis.

We thank the reviewer for highlighting this point, and have made the text clearer. We use clonal overlap analysis here instead of trajectory or pseudotime analysis for two reasons: (a) Whilst trajectory or pseudotime analysis can show where cells are currently differentiating, this cannot address the issue of how cells have differentiated prior to sampling. Indeed, the tumour aetiology in PDAC (and most cancers) has been shown to occur over years or decades¹², and therefore trajectory or pseudotime analysis will be limited in their insights into how the tumour microenvironment has been shaped over the tumour history. (b) The clonal sharing analysis is different from trajectory as this tells us how cells from the same clone (and thus antigen specificity) have differentiated within the whole history of each clone. Therefore, these analyses tell us that expanded CD8 T cell clones in ME patients are more likely to span the CD8 EM and CD8 senescent phenotypes. Taken the current knowledge of how T cells differentiate in a linear process it is still a valid conclusion to think of dysfunctional cells arising from previously activated cells.

We have now updated the text to reflect this point (line 292 in revised manuscript):

“Through quantifying the relative overlap of clones between different phenotypes within the CD4 and CD8 T cell populations, lineage patterns can be discerned (**Figure 3e**) over the history of the tumour prior to sampling. Indeed, the tumour aetiology in PDAC (and most cancers) has been shown to occur over years or decades¹², and clonal sharing analysis can determine how cells from the same clone (and thus antigen specificity) have differentiated within the whole history of each clone.”

14. In the “Immunosurveillance and resident tumour-infiltrating B and T cell clones are phenotypically distinct” section (lines 334-348), it’s important to know the peripheral abundance of the tumour-infiltrating clones they are analyzing. This would be particularly helpful to their hypothesis that certain immune cells may expand intratumorally, although it would be more robust to use bulk TCR/BCRseq of PBMCs to better sample the circulating repertoires (~100000s of immune receptors versus ~1000s of immune receptors with scRNAseq). If this isn’t feasible, reporting the abundance from the single-cell data could still be informative, as we anticipate more abundant clones to be better sampled by scRNA-seq.

We have now provided the number of immunosurveillance clones in Table S5 and all references are updated within the “Immunosurveillance and resident tumour-infiltrating B and T cell clones are phenotypically distinct” section.

We agree that bulk BCR or TCR sequencing would be useful to determine immunosurveillance to a greater sampling depth, and thus power. Unfortunately, the samples do not exist for bulk BCR or TCR sequencing on these patients, and the value of this data type would be limited to bulk cells, rather than being able to discriminate between phenotypes which our single cell analysis allows us to do.

We note, however, that we have a median of 287 immunosurveillance tumour cells per patient and 290 immunosurveillance blood cells per patient despite the limited cell capture afforded by single cell multi-omics suggests that the incidence of immunosurveillance B and T cells within PDAC patients is consistently being detected

15. “Finally, we show that only TIL T cell clones tend to be acutely activated with elevated with CD69 and PD1) whereas the blood counterparts of same clones are not acutely activated (Figure 4d).”, lines 416-419: Data regarding CD69 and PD1 expression in TIL T cell clones does not appear in Figure 4d.

Action taken:

Whilst figure 4d does show the upregulation of CD69 and PD1 (encoded by the PDCD1 gene) (on the right side of the figure panel), we clarify the text to make this clearer (line 409 in revised manuscript): “Finally, we show that only TIL T cell clones tend to be acutely activated with elevated with CD69 and PDCD1 (PD1) compared to the blood (Figure 4d).”

16. More methodologic details about sclsoTyper are needed (lines 698-701).

We have expanded on the details of sclsoTyper (from line 772 in revised manuscript):

“A pipeline, sclsoTyper, was written to assign most probable BCR IGH and IGK/L chains per droplet (based on nUMIs) and most probable TCR TRA and TRB chains per droplet (based on nUMIs). Briefly, sclsoTyper performs the following steps: (1) batches TCR and BCR data for IMGT submission; (2) parses the IMGT annotation results, for the quantification of SHM (using the IGHV identity output of IMGT), V/J gene usages, and CDR3 sequence identity; (3) identifies the highest expressed light chain and heavy chain BCR sequences per droplet, and the highest expressed alpha chain and beta chain TCR sequences per droplet; (3) clonality analysis between all samples from the same individual (allowing for assessment between blood and tumour). This is performed using a single-cell extension of established VDJ network construction software from Bashford-Rogers et al.²⁴ as part of sclsoTyper. A clone refers

to a group of clonally related B cells, each containing BCRs with identical CDR3 regions and IGHV gene use, or differing by single point mutations, such as through SHM. Each cluster is assumed to arise from the same pre-B cell. Clones were defined as a combination of (a) group of B cells expressing BCRs that are connected by SHM (i.e. within a cluster in a BCR network, where edges are generated between BCRs differing by one mutation), as previously described in 24, and (b) group of B cells expressing BCRs with identical CDR3 region DNA sequences and identical IGHV and IGHJ family identities. This definition allows for capturing clonal B cells expressing BCRs that have undergone SHM both within the CDR3 region and outside. (4) Finally, all this information is brought together in a meta-data format to include as part of the Seurat analysis and provides statistics on the number of droplets with heavy and/or light chains and TCR alpha and/or beta chains.”

17. For the cell-cell interaction analysis, were any comparisons done to other available methods? Either way, more methodologic detail rationalizing this approach is appropriate.

We thank the reviewer for raising this point and have expanded the methods section describing the cell-cell interaction analysis and its rational.

The reason for developing a cell-cell interaction methodology here was due to (a) the non-specific nature of the reference to immune cell signalling, with the abundance of false-positives and unrelated changes being observed, (b) the lack of normalisation between different samples, and (c) the inability to consider cell-intrinsic differences only, rather than considering cell number. The latter point is important as we have conceptually separated cell proportions (Figure 1) from cell-intrinsic differences (Figure 5) so as to distinguish these key aspects.

We have updated the methods section describing this analysis (from line 813 in revised manuscript): “Existing methods^{13,14} for analysing cell-cell interactions from scRNA-seq data suffer from (a) the non-specific nature of the reference to immune cell signalling, with the abundance of false-positives and unrelated changes being observed, (b) the lack of normalisation between different samples, and (c) the inability to consider cell-intrinsic differences only, rather than considering cell number. Here, we addressed all three aspects through a cell-cell communication tool. Firstly, reference receptor-ligand database was immune-cell-specific, generated through intercepting the human ligand-receptor database from Fantom (<https://fantom.gsc.riken.jp/>) with the genes that were captured in the PancrImmune scMulti-omics dataset (**Table S5**). Secondly, the signalling strengths between each pair of cell subtypes for each receptor-ligand pair was calculated by multiplying the percentage of cells per cell subtype expressing each respective gene for each patient. This was calculated for each cell subtype with ≥ 3 cells. Thus, the interaction strengths are independent of the total proportions of each cell type within the tumour or blood, and are normalised between each sample. This was plotted using *igraph* in R, and MANOVA was used to determine statistical differences between patient groups. The number of inbound and outbound links between cell subtypes was the counts of all corresponding non-zero receptor-ligand signalling strengths (**Figure 5b**). This was computed using *igraph* in R. Ranked interaction strengths per cell type were extracted per receptor-ligand pair for the Treg-specific analysis (**Figure 5c**).”

And updated the rational in the main text (from line 415 in revised manuscript):

“We have thus far shown that differential immune cell subtype frequencies distinguish ME and AE patients and lymphocyte-associated differences. We next examined the cell-intrinsic differences in cell-cell communication between immune cells between ME and AE patients. Here we considered cell-cell interaction strengths between known cytokine- and inflammation-associated ligands and their receptors (see **Methods**). Unlike in existing methods^{13,14}, we developed a workflow that considered only immune cell receptor-ligands, allowed for normalisation between samples, and examined cell-intrinsic differences only, rather than incorporating cell number. The signalling strengths between each

pair of cell subtypes for each receptor-ligand pair was calculated by multiplying the percentage of cells per cell subtype expressing each respective gene for each patient. Thus, the strengths are independent of the total proportions of each cell type within the tumour (Figure 5a).”

18. The following link does not work: <https://github.com/sakinaamin/BayesPrism> (line 785).

We apologise for not making the link public upon submission, and can confirm that this is now visible.

19. The manuscript lacks experimental validation. This is clearly outside the scope of this already large manuscript, but the findings should be clearly discussed as hypothesis-generating and not definitive. Softening the language, particularly in the abstract, introduction, and conclusions, would suffice, as well as clearly stating the need for experimental validation before considering intervention in patients.

Please see previous answers to reviewers. We have now added additional data to support our findings which will make the conclusions stronger. We have also tuned down the text where appropriate to reflect what is not a definitive experimental result.

Minor points:

1. Some acronyms have not been properly defined prior to their use in the text.

We have now updated the text to define acronyms:

- CD8 effector memory (EM)
- regulatory T cell (Treg)
- cytometry by time of flight (CyTOF)
- single cell RNAseq (gene expression, GEX)
- B cell receptor (BCR)
- T cell receptor (TCR)
- dendritic cells (DCs, known pAPCs)

2. Many typos, for example lines 99-100: “Importantly, we developed and applied novel single cell analyses to uncouple the distinct roles and contributions different immune cell populations...” should read “Importantly, we developed and applied novel single cell analyses to uncouple the distinct roles and contributions of different immune cell populations...”

We have now corrected the text accordingly.

3. Authors should avoid phrases like “the most detailed single-cell analysis” (lines 26-27).

We have now corrected the text accordingly to:

“We developed novel computational approaches applied to single-cell multi-omic data (scRNA-seq, CITE-seq, TCR-/BCR-seq) from matched tumour-infiltrated CD45+ cells and peripheral blood in 12 patients, generating a highly detailed single cell analysis of tumour-associated immune cells coupled with re-analysis of two publicly available datasets.”

4. In line 417: “...PD1) whereas the blood counterparts of same clones...”, the parentheses should be a comma.

We have now corrected the text accordingly.

References

- 1 Tempero, M. A. *et al.* Adjuvant nab-Paclitaxel + Gemcitabine in Resected Pancreatic Ductal Adenocarcinoma: Results From a Randomized, Open-Label, Phase III Trial. *J Clin Oncol* **41**, 2007-2019 (2023). <https://doi.org:10.1200/JCO.22.01134>
- 2 Nguyen, V. & Griss, J. scAnnotatR: framework to accurately classify cell types in single-cell RNA-sequencing data. *BMC Bioinformatics* **23**, 44 (2022). <https://doi.org:10.1186/s12859-022-04574-5>
- 3 Aran, D. *et al.* Reference-based analysis of lung single-cell sequencing reveals a transitional profibrotic macrophage. *Nat Immunol* **20**, 163-172 (2019). <https://doi.org:10.1038/s41590-018-0276-y>
- 4 Lyu, P., Zhai, Y., Li, T. & Qian, J. CellAnn: a comprehensive, super-fast, and user-friendly single-cell annotation web server. *Bioinformatics* **39** (2023). <https://doi.org:10.1093/bioinformatics/btad521>
- 5 Ianevski, A., Giri, A. K. & Aittokallio, T. Fully-automated and ultra-fast cell-type identification using specific marker combinations from single-cell transcriptomic data. *Nat Commun* **13**, 1246 (2022). <https://doi.org:10.1038/s41467-022-28803-w>
- 6 Yang, L. *et al.* Single-cell Mayo Map (scMayoMap): an easy-to-use tool for cell type annotation in single-cell RNA-sequencing data analysis. *bioRxiv* (2023). <https://doi.org:10.1101/2023.05.03.538463>
- 7 Hao, Y. *et al.* Integrated analysis of multimodal single-cell data. *Cell* **184**, 3573-3587 e3529 (2021). <https://doi.org:10.1016/j.cell.2021.04.048>
- 8 Dominguez Conde, C. *et al.* Cross-tissue immune cell analysis reveals tissue-specific features in humans. *Science* **376**, eabl5197 (2022). <https://doi.org:10.1126/science.abl5197>
- 9 Peng, J. *et al.* Single-cell RNA-seq highlights intra-tumoral heterogeneity and malignant progression in pancreatic ductal adenocarcinoma. *Cell Res* **29**, 725-738 (2019). <https://doi.org:10.1038/s41422-019-0195-y>
- 10 Steele, N. G. *et al.* Multimodal Mapping of the Tumor and Peripheral Blood Immune Landscape in Human Pancreatic Cancer. *Nat Cancer* **1**, 1097-1112 (2020). <https://doi.org:10.1038/s43018-020-00121-4>
- 11 Sun, W., Chang, C. & Long, Q. Bayesian Non-linear Support Vector Machine for High-Dimensional Data with Incorporation of Graph Information on Features. *Proc IEEE Int Conf Big Data* **2019**, 4874-4882 (2019). <https://doi.org:10.1109/bigdata47090.2019.9006473>
- 12 Yu, J., Blackford, A. L., Dal Molin, M., Wolfgang, C. L. & Goggins, M. Time to progression of pancreatic ductal adenocarcinoma from low-to-high tumour stages. *Gut* **64**, 1783-1789 (2015). <https://doi.org:10.1136/gutjnl-2014-308653>
- 13 Jin, S. *et al.* Inference and analysis of cell-cell communication using CellChat. *Nat Commun* **12**, 1088 (2021). <https://doi.org:10.1038/s41467-021-21246-9>
- 14 Efremova, M., Vento-Tormo, M., Teichmann, S. A. & Vento-Tormo, R. CellPhoneDB: inferring cell-cell communication from combined expression of multi-subunit ligand-receptor complexes. *Nat Protoc* **15**, 1484-1506 (2020). <https://doi.org:10.1038/s41596-020-0292-x>

REVIEWERS' COMMENTS

Reviewer #1 (Remarks to the Author):

The authors adequately addressed the major concern of the manuscript.

Minor comment:

In the new data the authors performed IHQ studies. In the macrophage analysis they stained for CD163+ CMAF+ which identifies M2 macrophages. Authors should mention that detected macrophages were of M2 population.

Reviewer #2 (Remarks to the Author):

All my prior concerns have been addressed. The inclusion of new data in support of the findings strengthens the conclusion.

Reviewer #3 (Remarks to the Author):

Many of the concerns are sufficiently addressed. Further supportive data is provided and drawn from public available data demonstrating correlations. Definitive experimental validation is still lacking but could be reasonably addressed in a follow up manuscript.

Dear reviewers, we would like to thank you for reviewing the manuscript again and we thank you helping us improve the manuscript

Reviewer #1 (Remarks to the Author):

The authors adequately addressed the major concern of the manuscript.

Thank you

Minor comment:

In the new data the authors performed IHQ studies. In the macrophage analysis they stained for CD163+ CMAF+ which identifies M2 macrophages. Authors should mention that detected macrophages were of M2 population.

We have modified the manuscript to acknowledge this important point.

Reviewer #2 (Remarks to the Author):

All my prior concerns have been addressed. The inclusion of new data in support of the findings strengthens the conclusion.

Thank you for your comments

Reviewer #3 (Remarks to the Author):

Many of the concerns are sufficiently addressed. Further supportive data is provided and drawn from public available data demonstrating correlations. Definitive experimental validation is still lacking but could be reasonably addressed in a follow up manuscript.

Thank you for your comments and we will be pursuing further experimental validation that can hopefully go into another manuscript.